# Activation of mitochondrial unfolded protein response protects against multiple exogenous stressors

Sonja K Soo[1,2], Annika Traa[1,2] , Paige D Rudich[1,2] , Meeta Mistry[3], Jeremy M Van Raamsdonk[1,2,4,5]

The mitochondrial unfolded protein response (mitoUPR) is an evolutionarily conserved pathway that responds to mitochondria insults through transcriptional changes, mediated by the transcription factor ATFS-1/ATF-5, which acts to restore mitochondrial homeostasis. In this work, we characterized the role of ATFS-1 in responding to organismal stress. We found that activation of ATFS-1 is sufficient to cause up-regulation of genes involved in multiple stress response pathways including the DAF-16–mediated stress response pathway, the cytosolic unfolded protein response, the endoplasmic reticulum unfolded protein response, the SKN-1–mediated oxidative stress response pathway, the HIF-1-mediated hypoxia response pathway, the p38-mediated innate immune response pathway, and antioxidant genes. Constitutive activation of ATFS-1 increases resistance to multiple acute exogenous stressors, whereas disruption of *atfs-1* decreases stress resistance. Although ATFS-1–dependent genes are up-regulated in multiple long-lived mutants, constitutive activation of ATFS-1 decreases lifespan in wild-type animals. Overall, our work demonstrates that ATFS-1 serves a vital role in organismal survival of acute stressors through its ability to activate multiple stress response pathways but that chronic ATFS-1 activation is detrimental for longevity.

## Introduction

The mitochondrial unfolded protein response (mitoUPR) is a stress response pathway that acts to reestablish mitochondrial homeostasis by inducing transcriptional changes in genes involved in the metabolism and restoration of mitochondrial protein folding (Zhao et al, 2002). Various perturbations to the mitochondria can activate mitoUPR, including disruption of mitochondrial translation, disruption of mitochondrial protein synthesis, impairment of oxidative phosphorylation, disruption of mitochondrial proteostasis, altered metabolism, defects in mitochondrial DNA, excess reactive oxygen species (ROS), disruption of protein degradation, and defects in mitochondrial import (Shpilka & Haynes, 2018). The mitoUPR is mediated by the transcription factor activating transcription factor associated with stress-1 (ATFS-1) in *Caenorhabditis elegans* (Nargund et al, 2012), and activating transcription factor 5 (ATF5) in mammals (Fiorese et al, 2016).

ATFS-1/ATF5 regulates the mitoUPR through its dual targeting domains: a mitochondrial targeting sequence (MTS) and a NLS. Under normal unstressed conditions, the MTS causes ATFS-1 to enter the mitochondria through the HAF-1 import channel. Inside the mitochondria, ATFS-1 is degraded by the protease CLPP-1/CLP1 (Nargund et al, 2012). However, when mitochondrial import or degradation of ATFS-1 is disrupted under conditions of mitochondrial stress, ATFS-1 accumulates in the cytoplasm. The NLS of the cytoplasmic ATFS-1 then targets it to the nucleus, where ATFS-1 acts with the transcription factor DVE-1 and transcriptional regulator UBL-5 to up-regulate expression of chaperones, proteases, and other proteins (Jovaisaite et al, 2014).

In order to study the role of the mitoUPR in longevity, we previously disrupted *atfs-1* in long-lived *nuo-6* mutants, which contain a point mutation that affects complex I of the electron transport chain (Yang & Hekimi, 2010b). *nuo-6* mutants have a mild impairment of mitochondrial function that leads to increased lifespan and enhanced resistance to multiple stressors. We found that loss of *atfs-1* not only decreased the lifespan of *nuo-6* worms but also abolished the increased stress resistance of these worms, thereby suggesting that ATFS-1 contributes to both longevity and stress resistance in these worms (Wu et al, 2018).

Although a role for the mitoUPR in longevity has been reported (Durieux et al, 2011; Houtkooper et al, 2013; Berendzen et al, 2016; Merkwirth et al, 2016) and debated (Bennett et al, 2014; Bennett & Kaeberlein, 2014), little is known about the role of ATFS-1 in response to exogenous stressors. Activation of ATFS-1 can increase organismal resistance to the pathogenic bacterium *Pseudomonas aeruginosa* (Pellegrino et al, 2014) and can protect against anoxia–reperfusion–induced death (Pena et al, 2016).

In this study, we use *C. elegans* to define the relationship between ATFS-1 and organismal stress resistance and to explore the

[1]Department of Neurology and Neurosurgery, McGill University, Montreal, Canada [2]Metabolic Disorders and Complications Program, and Brain Repair and Integrative Neuroscience Program, Research Institute of the McGill University Health Centre, Montreal, Canada [3]Bioinformatics Core, Harvard School of Public Health, Harvard Medical School, Boston, MA, USA [4]Division of Experimental Medicine, Department of Medicine, McGill University, Montreal, Canada [5]Department of Genetics, Harvard Medical School, Boston, MA, USA

Correspondence: jeremy.vanraamsdonk@mcgill.ca

underlying mechanisms. We find that activation of ATFS-1 is sufficient to up-regulate genes from multiple stress response pathways and is important for transcriptional changes induced by oxidative stress and bacterial pathogen exposure. Constitutive activation of ATFS-1 is also sufficient to increase resistance to multiple external stressors. Although ATFS-1–dependent genes are up-regulated in several long-lived mutants that are representative of multiple pathways of lifespan extension, chronic activation of ATFS-1 does not extend longevity. Overall, our results demonstrate a crucial role for ATFS-1 in organismal stress response through activation of multiple stress response pathways.

# Results

## ATFS-1 activates genes from multiple stress response pathways

Mild impairment of mitochondrial function by a mutation in *nuo-6* results in the activation of the mitoUPR. We previously performed a bioinformatics analysis of genes that are up-regulated in *nuo-6* mutants in an ATFS-1–dependent manner and discovered an enrichment of genes associated with the GO term "response to stress" (Wu et al, 2018). Based on this observation, we hypothesized that ATFS-1 may be able to activate other stress response pathways. To test this hypothesis, we quantified the expression of established target genes from eight different stress response pathways under conditions where ATFS-1 is either activated or disrupted.

For this analysis, we picked target genes that have been commonly used in the literature to represent their associated stress response pathway. These target genes included *hsp-6* in the mitochondrial unfolded protein response (mitoUPR) pathway (Yoneda et al, 2004; Dues et al, 2016); *hsp-4* in the ER unfolded protein response (ER-UPR) pathway (Urano et al, 2002; Dues et al, 2016); *hsp-16.2* in the cytoplasmic unfolded protein response pathway (cytoUPR) (Link et al, 1999; Dues et al, 2016); *sod-3* in the DAF-16–mediated stress response pathway (Honda & Honda, 1999; Dues et al, 2016); *gst-4* in the SKN-1–mediated stress response pathway (Kahn et al, 2008; Dues et al, 2016); *nhr-57* in the HIF-1–mediated hypoxia response pathway (Bishop et al, 2004; Dues et al, 2016); *Y9C9A.8* in the p38-mediated innate immunity pathway (Fletcher et al, 2019; Campos et al, 2021); and *trx-2*, an antioxidant gene (Cacho-Valadez et al, 2012) (Table S1).

To activate ATFS-1, we used the *nuo-6* mutation. We also examined gene expression in two different gain-of-function (GOF) mutants with constitutively active ATFS-1: *atfs-1(et15)* and *atfs-1(et17)*. Both of these constitutively active ATFS-1 mutants have mutations in the MTS which increase nuclear localization of ATFS-1 (Rauthan et al, 2013). To identify ATFS-1–dependent genes, we used a loss-of-function (LOF) *atfs-1* deletion mutation (*gk3094*) to disrupt ATFS-1 function in wild-type worms and *nuo-6* mutants.

We found that compared with wild-type worms, *atfs-1(gk3094)* deletion mutants did not have decreased expression levels for the target genes of any of the stress response pathways (Fig 1). This indicates that ATFS-1 is not required for the basal expression levels of these stress response genes.

Activation of the mitoUPR through mutation of *nuo-6* resulted in significant up-regulation of target genes from the mitoUPR (*hsp-6*;

Fig 1A), the DAF-16–mediated stress response (*sod-3*; Fig 1D), the SKN-1–mediated oxidative stress response (*gst-4*; Fig 1E), the HIF-1–mediated hypoxia response (*nhr-57*; Fig 1F), the p38-mediated innate immunity pathway (*Y9C9A.8*; Fig 1G), and antioxidant defense (*trx-2*; Fig 1H). Importantly, for all of these genes, inhibiting the mitoUPR through disruption of *atfs-1* prevented the up-regulation of the stress response in *nuo-6;atfs-1(gk3094)* worms (Fig 1A and D–H), indicating that ATFS-1 is required for the activation of these stress pathway genes during mitochondrial stress.

Constitutive activation of ATFS-1 in *atfs-1(et15)* mutants resulted in up-regulation of the majority of the target genes up-regulated in *nuo-6* mutants, except for the SKN-1 target gene *gst-4* (Fig 1A and D–H). Similarly, constitutively active *atfs-1(et17)* mutants result in significant up-regulation of *hsp-6*, *sod-3*, *Y9C9A.8*, and *trx-2* and a nonsignificant 77% increase in *nhr-57* expression (Fig 1). This indicates that ATFS-1 activation is sufficient to induce up-regulation of specific stress response genes independent of mitochondrial stress. Activating the mitoUPR through the *nuo-6* mutation or through the constitutively active ATFS-1 mutants did not significantly increase the expression of the ER-UPR target gene *hsp-4* (Fig 1B) or the cyto-UPR target gene *hsp-16.2* (Fig 1C). However, both the *nuo-6* mutant and the constitutively active ATFS-1 mutants had a 2.5- to 19.5-fold increase in *hsp-16.2* levels, which failed to reach significance due to variability between replicates, and the fact that *hsp-16.2* expression levels can be increased up to 60-fold.

As only one gene was examined per stress response pathway, it is possible that different target genes may yield a different result. In addition, some of the stress response genes that we examined are not exclusively activated by the pathway that they are frequently used to represent. For example, *gst-4* is an antioxidant gene that is commonly used as a readout of SKN-1 activity but can also be activated by DAF-16 and the mitoUPR (see Table S3 for lists of genes that are up-regulated by activation of different stress response pathways).

To circumvent these potential limitations and to gain a more comprehensive view of the extent to which mitoUPR activation causes up-regulation of genes in other stress response pathways, we compared genes up-regulated in the constitutively active *atfs-1* mutant, *atfs-1(et15)*, with genes up-regulated by activation of different stress response pathways. As a proof of principle, we first examined the overlap between up-regulated genes in *atfs-1(et15)* mutants and genes up-regulated by activation of the mitoUPR with *spg-7* RNAi in an ATFS-1–dependent manner (Nargund et al, 2012).

We identified genes up-regulated by the activation of other stress response pathways from published gene expression studies. The genes and relevant pathways are listed in Table S3. ER-UPR pathway target genes were defined as genes up-regulated by tunicamycin exposure and dependent on *ire-1*, *xbp-1*, *pek-1*, or *atf-6* (Shen et al, 2005b). Cyto-UPR pathway genes are genes up-regulated by overexpression of heat shock factor 1 (HSF-1) and genes bound by HSF-1 after a 30-min heat shock at 34°C (Li et al, 2016; Sural et al, 2019). DAF-16 pathway genes were identified by Tepper et al by performing a meta-analysis of 46 previous gene expression studies, comparing conditions in which DAF-16 is activated (e.g., *daf-2* mutants) and conditions in which the activation is inhibited by disruption of *daf-16* (e.g., *daf-2;daf-16* mutants) (Tepper et al, 2013). SKN-1 pathway genes were identified as genes that exhibit decreased expression in wild-type worms treated with

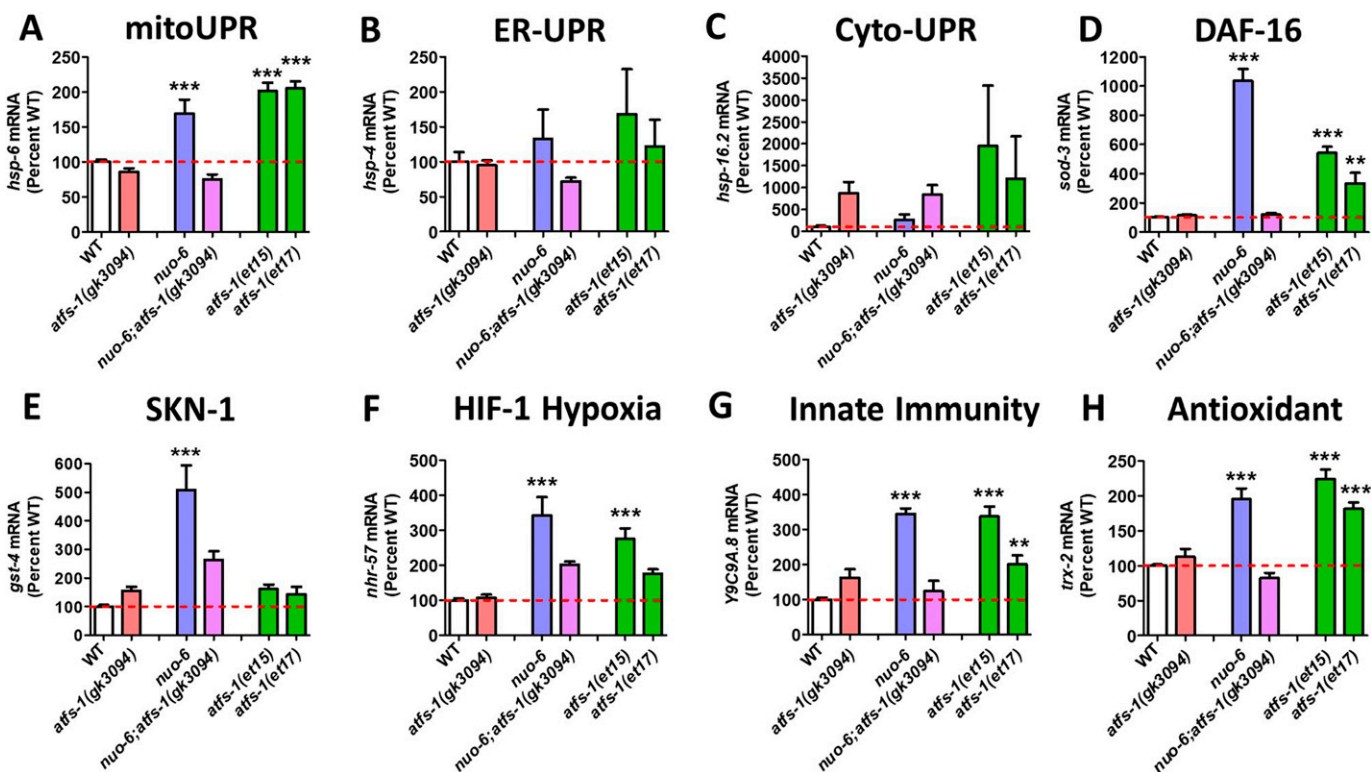

**Figure 1. Activation of ATFS-1 up-regulates genes from multiple stress response pathways.**
To determine the role of ATFS-1 in the activation of genes from different stress response pathways, we activated ATFS-1 by mildly impairing mitochondrial function through a mutation in *nuo-6* (blue bars) and then examined the effect of disrupting *atfs-1* using an *atfs-1* deletion mutant *atfs-1(gk3094)* (purple bars). We also examined the expression of these genes in two constitutively active *atfs-1* mutants, *atfs-1(et15)* and *atfs-1(et17)* (green bars). **(A, B, C, D, E, F, G, H)** Target genes from the mitochondrial unfolded protein response (A, mitoUPR, *hsp-6*), the ER unfolded protein response (B, ER-UPR, *hsp-4*), the cytoplasmic unfolded protein response (C, Cyto-UPR, *hsp-16.2*), the DAF-16–mediated stress response (D, *sod-3*), SKN-1–mediated oxidative stress response (E, *gst-4*), HIF-1–mediated hypoxia response (F, *nhr-57*), p38-mediated innate immune pathway (G, *Y9C9A.8*), and antioxidant defense (H, *trx-2*) were measured. Target genes from the mitoUPR, DAF-16–mediated stress response, SKN-1–mediated oxidative stress response, HIF-1–mediated hypoxia response, p38-mediated innate immune pathway, and antioxidant defense are all significantly up-regulated in *nuo-6* mutants in an ATFS-1–dependent manner. Target genes from the mitoUPR, DAF-16–mediated stress response, HIF-1–mediated hypoxia response, p38-mediated innate immune pathway, and antioxidant defense are also up-regulated in at least one of the constitutively activated *atfs-1* mutants. In contrast, activation of ATFS-1 by *nuo-6* mutation or *atfs-1* gain-of-function mutations did not significantly affect target gene expression for the ER-UPR or the Cyto-UPR. *atfs-1(gk3094)* is a loss-of-function deletion mutant. *atfs-1(et15)* and *atfs-1(et17)* are constitutively active gain-of-function mutants. A full list of genes that are up-regulated by ATFS-1 activation can be found in Table S2. Data information: Error bars indicate SEM. **P < 0.01, ***P < 0.001. Statistical analysis was performed using a one-way ANOVA with the Bonferroni post hoc test. The number of replicates and statistical analysis can be found in Table S6.

*skn-1* RNAi, genes that are up-regulated in *glp-1* mutants in an SKN-1–dependent manner, genes that are up-regulated by germ line stem cell removal in an SKN-1–dependent manner (Steinbaugh et al, 2015), and genes up-regulated in *daf-2* mutants in an SKN-1–dependent manner (Ewald et al, 2015). HIF-1–mediated hypoxia genes are genes induced by hypoxia in an HIF-1–dependent manner (Shen et al, 2005a). Innate immunity genes are defined as genes up-regulated by exposure to *P. aeruginosa* strain PA14 in a PMK-1– and ATF-7–dependent manner (Fletcher et al, 2019), where PMK-1 and ATF-7 are part of the p38-mediated innate immune signaling pathway. Finally, antioxidant genes include a comprehensive list of genes involved in antioxidant defense such as superoxide dismutases (*sod*), catalases (*ctl*), peroxiredoxins (*prdx*), or thioredoxins (*trx*).

In comparing genes up-regulated in the constitutively active *atfs-1* mutant *et15* with the previously published gene lists, we found that 51% of genes up-regulated by *spg-7* RNAi in an ATFS-1–dependent manner are also up-regulated by constitutive activation of ATFS-1 (Fig 2A). Similarly, there was a highly significant

overlap of up-regulated genes between *atfs-1(et15)* mutants and each of the other stress response pathways. *atfs-1(et15)* had a 25% overlap with genes of the ER-UPR pathway (Fig 2B); 22% overlap with genes of the Cyto-UPR pathway (Fig 2C); 26% overlap with genes of the DAF-16–mediated stress response pathway (Fig 2D); 30% overlap with genes of the SKN-1–mediated oxidative stress response pathway (Fig 2E); 23% overlap with genes of the HIF-1–mediated hypoxia response pathway (Fig 2F); 22% overlap with genes of the p38-mediated innate immunity pathway (Fig 2G); and 33% overlap with antioxidant genes (Fig 2H). Combined, this indicates that activation of ATFS-1 is sufficient to up-regulate target genes in multiple stress response pathways.

To determine the extent to which genes common to multiple stress response pathways are up-regulated by ATFS-1 activation, we generated an UpSetR plot to simultaneously compare the overlaps between all of these gene sets. We found that there are many genes that can be up-regulated by activation of different stress response pathways (Figs 2I and S1A and B and Table S4). In addition, there are

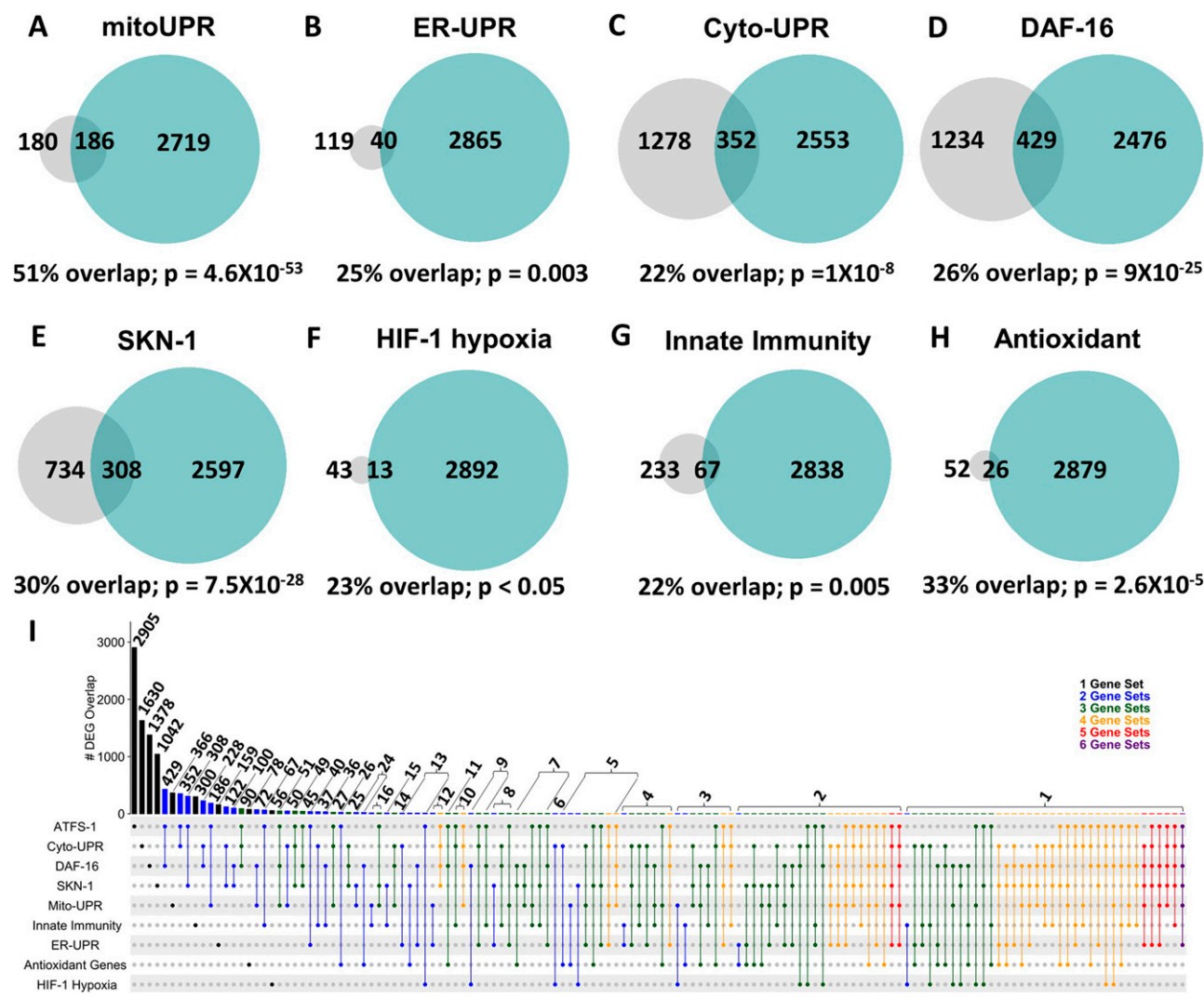

**Figure 2. Constitutive activation of ATFS-1 results in up-regulation of genes from multiple stress response pathways.**
**(A, B, C, D, E, F, G, H)** Genes that are up-regulated by constitutive activation of ATFS-1 were compared with previously published lists of genes involved in different stress response pathways, including the mitochondrial unfolded protein response (A, mitoUPR), the ER unfolded protein response (B, ER-UPR), the cytoplasmic unfolded protein response (C, Cyto-UPR), the DAF-16–mediated stress response (D), the SKN-1–mediated oxidative stress response (E), the HIF-1–mediated hypoxia response (F), the p38-mediated innate immune response (G), and antioxidant genes (H). In every case, there was a significant degree of overlap ranging from 22 to 51%. Grey circles indicate genes that are up-regulated by activation of the stress response pathway indicated. Turquoise circles indicate genes that are up-regulated in the *atfs-1(et15)* constitutively active gain-of-function mutant. The numbers inside the circles show how many genes are up-regulated. The percentage overlap is the number of overlapping genes as a percentage of the number of genes up-regulated by the stress response pathway. *P*-values indicate the significance of the difference between the observed number of overlapping genes between the two gene sets, and the expected number of overlapping genes if the genes were picked at random. Panel **(I)** shows an inclusive UpSetR plot displaying the overlap between up-regulated genes associated with each stress response pathway. Vertical bars indicate the number of genes in common (overlap) between gene sets indicated by the dots below. Horizontal black bars indicate the number of genes within each gene set. mitoUPR, mitochondrial unfolded protein response; ER-UPR, endoplasmic reticulum unfolded protein response; Cyto-UPR, cytoplasmic unfolded protein response; DAF-16, DAF-16–mediated stress response pathway; SKN-1, SKN-1–mediated oxidative stress response pathway; HIF-1, HIF-1–mediated hypoxia response pathway; innate immunity, p38-mediated innate immunity pathway; antioxidant, antioxidant genes. Stress pathway gene lists and sources can be found in Table S3. Lists of genes common to multiple stress response pathways can be found in Table S4.

multiple genes that are up-regulated by ATFS-1 activation and independent of other stress response pathways (Table S4).

### ATFS-1 can bind to the same promoter as other stress-responsive transcription factors

The fact that ATFS-1 activation results in the up-regulation of the same genes as activation of other stress response pathways does not imply direct regulation of these genes by either transcription factor. ATFS-1 could modulate these genes either directly by binding to promoter or enhancer elements, or indirectly by acting on other transcription factors or altering metabolism or physiology. To gain insights into the mechanism of regulation, we sought to determine if ATFS-1 can bind to the same genes as other stress-responsive transcription factors. We compared previously published chromatin immunoprecipitation sequencing (ChIP-seq)

experiments involving ATFS-1 (Nargund et al, 2015), HSF-1 (Kovacs et al, 2019), DAF-16 (Kumar et al, 2015; Webb et al, 2016), SKN-1 (Niu et al, 2011), HIF-1 (Kudron et al, 2018), and ATF-7 (Fletcher et al, 2019).

We found that ATFS-1 can bind to several of the same genes as other stress-responsive transcription factors (Fig S2). The degree of overlap ranged from 16% for HSF-1 to 61% for HIF-1. This suggests that ATFS-1 can directly regulate these genes. However, indirect regulation of gene expression could also contribute to the overlap in gene expression observed in Fig 2. It is important to note that these ChIP-seq experiments were performed under different conditions (e.g., ATFS-1 was examined in response to *spg-7* RNAi and ATF-7 was examined in response to bacterial pathogen exposure). If these experiments were performed under the same conditions, the degree of overlap could be different from that under these specific conditions.

## ATFS-1 is required for transcriptional responses to exogenous stressors

Having shown that constitutive activation of ATFS-1 can induce up-regulation of genes involved in various stress response pathways, we next sought to determine the role of ATFS-1 in the genetic response to different stressors. To do this, we exposed wild-type animals and *atfs-1(gk3094)* LOF mutants to six different external stressors and quantified the resulting up-regulation of stress response genes using quantitative RT-PCR (qPCR). The examined stress response genes were the established target genes of the stress response pathways that we examined in Fig 1 and genes that we previously identified as up-regulated by specific stressors using fluorescent reporter strains (Dues et al, 2016). These genes included *hsp-6, hsp-4, hsp-16.2, sod-3, gst-4, nhr-57, Y9C9A.8, trx-2, ckb-2, gcs-1, sod-5, T24B8.5/sysm-1, clec-67,* and *dod-22.* We found that exposure to either oxidative stress (4 mM paraquat, 48 h) or the bacterial pathogen *P. aeruginosa* strain PA14 induced a significant up-regulation of stress response genes in wild-type worms, which was suppressed by disruption of *atfs-1* (Figs 3A and B, S3, and S4). In contrast, exposure to heat stress (35°C, 2 h; Figs 3C and S5), osmotic stress (300 mM NaCl, 24 h; Figs 3D and S6), anoxic stress (24 h; Figs 3E and S7), or ER stress (tunicamycin for 24 h; Figs 3F and S8) caused up-regulation of stress response genes in both wild-type and *atfs-1(gk3094)* worms to a similar extent, or to a greater extent in *atfs-1* deletion mutants. Combined, these results indicate that ATFS-1 is required for up-regulating stress response genes in response to exposure to oxidative stress or bacterial pathogens. Although we did not observe evidence for a role of ATFS-1 in up-regulating stress response genes following exposure to other stressors, it is possible that there are genes that we did not examine that are up-regulated by the other four stressors in an ATFS-1–dependent manner.

## Modulation of ATFS-1 levels affects resistance to multiple stressors

Due to the crucial role of ATFS-1 in up-regulating genes in multiple stress response pathways, we next sought to determine the extent to which activating ATFS-1 protects against exogenous stressors. We quantified resistance to stress in two constitutively active *atfs-1* GOF mutants (*atfs-1(et15), atfs-1(et17)*) compared with wild-type

worms. For comparison, we also included an *atfs-1* LOF deletion mutant (*atfs-1(gk3094)*), which we previously found to have decreased resistance to oxidative stress, heat stress, osmotic stress, and anoxic stress (Wu et al, 2018).

Resistance to acute oxidative stress was measured by exposing worms to 300 µM juglone. We found that both GOF mutants, *atfs-1(et15)* and *atfs-1(et17)*, have increased resistance to acute oxidative stress compared with wild-type worms, while *atfs-1(gk3094)* deletion mutants were less resistant compared to wild-type worms (Fig 4A). To quantify resistance to chronic oxidative stress, worms were transferred to plates containing 4 mM paraquat beginning at day 1 of adulthood until death. Similar to the acute assay, *atfs-1(et17)* mutants were more resistant to chronic oxidative stress, whereas *atfs-1(gk3094)* mutants were less resistant to chronic oxidative stress compared to wild-type worms (Fig 4B). Oddly, *atfs-1(et15)* GOF mutants exhibited decreased resistance to chronic oxidative stress. The diminished protection in *atfs-1(et17)* mutants and lack of protection in the *atfs-1(et15)* mutants in the paraquat assay may be due to the chronic nature of the assay, compared with the juglone assay which measures resistance to acute oxidative stress.

Resistance to heat stress was measured by incubating worms at 37°C. None of the mutants showed increased survival during heat stress, with both *atfs-1(et15)* and *atfs-1(gk3094)* mutants exhibiting a significant decrease in survival compared with wild-type worms (Fig 4C). Resistance to ER stress was measured by exposing worms to 50 µg/ml tunicamycin. We found that *atfs-1(et15)* and *atfs-1(et17)* constitutively active mutants have increased resistance to ER stress, whereas *atfs-1(gk3094)* deletion mutants have an equivalent survival to wild-type worms (Fig 4D). Resistance to osmotic stress was quantified on plates containing 500 mM NaCl after 48 h. Under these conditions, the constitutively active *atfs-1* mutants had increased survival compared with wild-type worms, whereas *atfs-1(gk3094)* deletion mutants had decreased survival (Fig 4E). Resistance to anoxic stress was measured by placing worms in an oxygen-free environment for 75 h, followed by a 24-h recovery period. We observed increased survival in *atfs-1(et15)* and *atfs-1(et17)* mutants and a trend towards decreased survival in *atfs-1(gk3094)* mutant compared with wild-type worms (Fig 4F).

Lastly, to test resistance to bacterial pathogens, worms were exposed to *P. aeruginosa* strain PA14 in either a fast kill assay, in which worms die from a toxin produced by the bacteria, or a slow kill assay, in which worms die due to the intestinal colonization of the pathogenic bacteria (Kirienko et al, 2014). In the fast kill assay, constitutive activation of ATFS-1 increased survival in *atfs-1(et15)* and *atfs-1(et17)* mutants compared with wild-type worms (Fig 4G). *atfs-1(gk3094)* deletion mutants also exhibited increased survival. For the slow kill assay, we used two established protocols: one in which the assay is initiated at the L4 larval stage and performed at 25°C (Kirienko et al, 2014; Pellegrino et al, 2014; Dues et al, 2016) and the other in which the assay is initiated at day 3 of adulthood and performed at 20°C (Wu et al, 2019). Surprisingly, at 25°C, we found that the *atfs-1(et17)* mutant had a small decrease in resistance to PA14, whereas *atfs-1(gk3094)* mutants exhibited a small increase in resistance to PA14 compared with wild-type worms (Fig 4H). At 20°C, both *atfs-1(gk3094)* and *atfs-1(et17)* mutants had a small increase in resistance to PA14 compared with wild-type worms (Fig 4I).

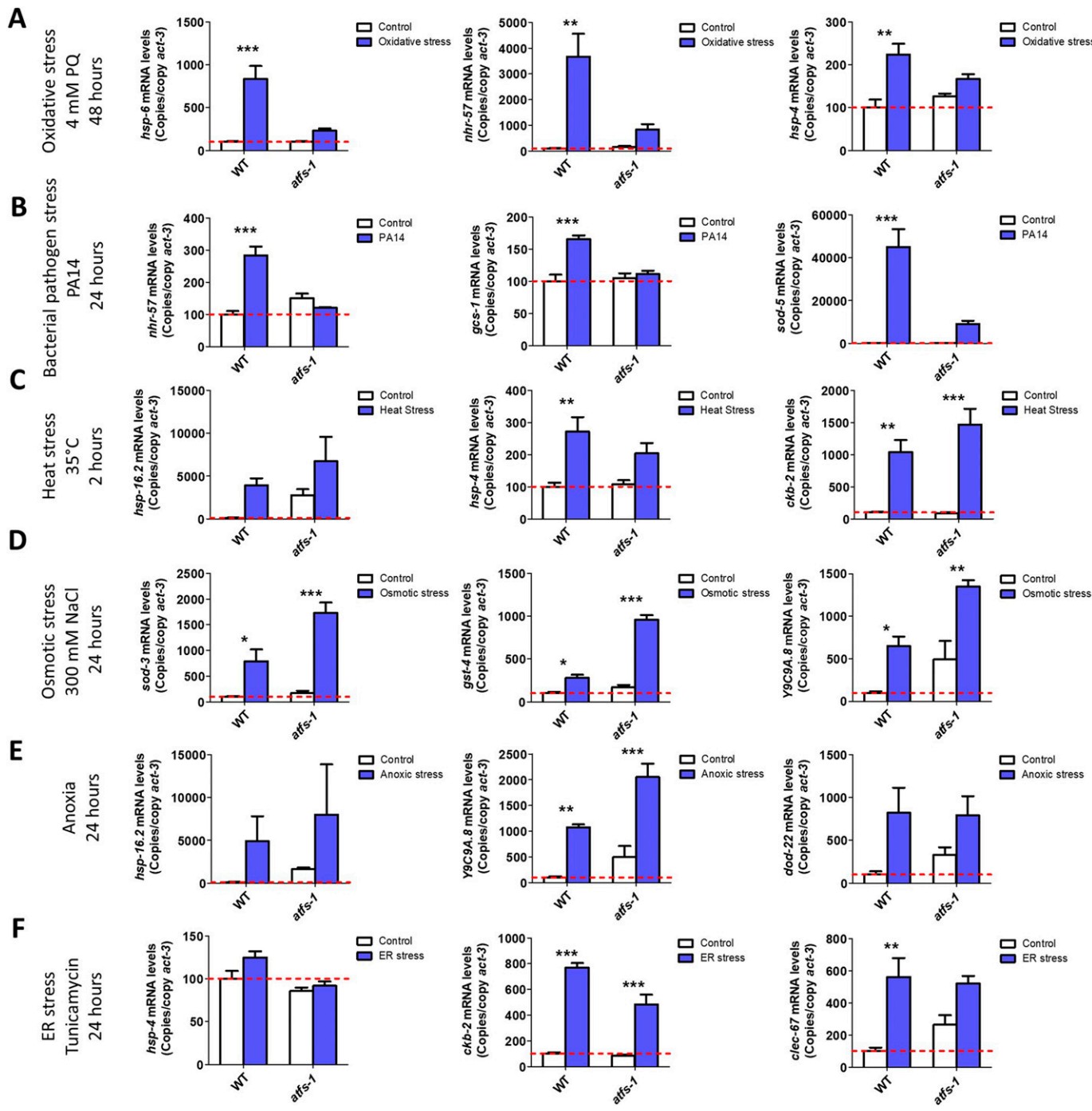

**Figure 3. ATFS-1 is required for up-regulation of stress response genes after exposure to oxidative stress or bacterial pathogen stress.**
To determine the role of ATFS-1 in responding to different types of stress, we compared the up-regulation of stress response genes in wild-type and *atfs-1(gk3094)* loss-of-function deletion mutants after exposure to different stressors. **(A)** Exposure to oxidative stress (4 mM paraquat, 48 h) caused a significant up-regulation of *hsp-6*, *nhr-57*, and *trx-2* in wild-type worms that was prevented by the disruption of *atfs-1*. **(B)** Exposure to bacterial pathogen stress (PA14, 24 h) resulted in an up-regulation of *nhr-57*, *gcs-1* and *sod-5* in wild-type worms that was prevented by the *atfs-1* deletion. **(C)** Exposure to heat stress (35°C, 2 h) caused increased expression of *ckb-2* and a trend towards increased expression of *hsp-16.2* and *hsp-4* in both wild-type and *atfs-1* worms. **(D)** Exposure to osmotic stress (300 mM, 24 h) caused an up-regulation of *sod-3*, *gst-4*, and *Y9C9A.8* in wild-type worms and to a greater magnitude in *atfs-1* mutants. **(E)** Anoxia (24 h) resulted in the up-regulation of *hsp-16.2*, *Y9C9A.8*, and *dod-22* in both wild-type and *atfs-1* worms. **(F)** Exposing worms to ER stress (5 μg/ml tunicamycin, 24 h) increased the expression of *ckb-2* and trended towards increasing the expression of *clec-67* in both wild-type and *atfs-1* worms. Data information: Error bars indicate SEM. *$P < 0.05$, **$P < 0.01$, ***$P < 0.001$. Statistical analysis was performed using a two-way ANOVA with a Bonferroni post hoc test. The number of replicates and statistical analysis can be found in Table S6.

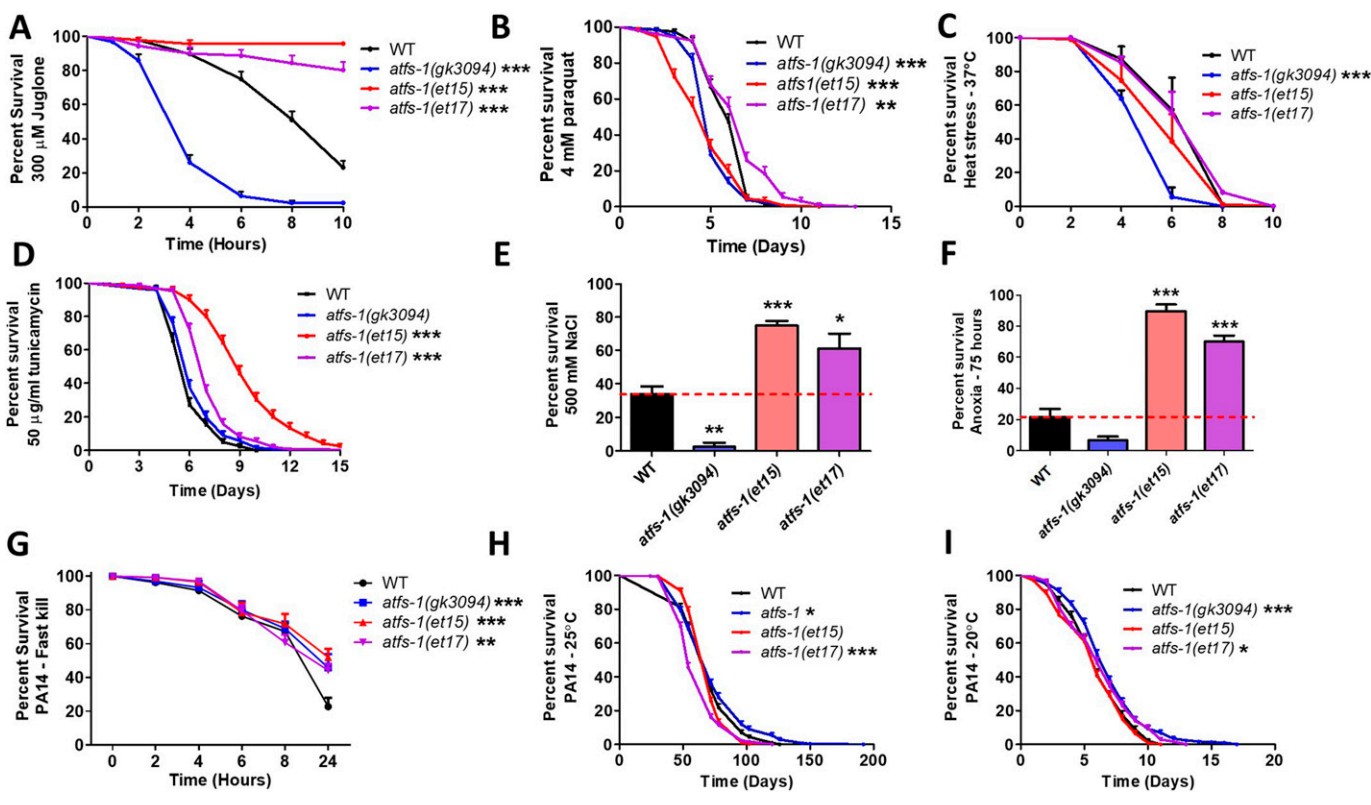

**Figure 4. Constitutive activation of ATFS-1 increases resistance to multiple external stressors.**
To determine the role of ATFS-1 in resistance to stress, the stress resistance of an *atfs-1* loss-of-function mutants (*atfs-1(gk3094)*) and two constitutively active *atfs-1* gain-of-function mutants (*atfs-1(et15)*, *atfs-1(et17)*) was compared with wild-type worms. **(A)** Activation of ATFS-1 enhanced resistance to acute oxidative stress (300 μM juglone), whereas disruption of *atfs-1* markedly decreased resistance to acute oxidative stress. **(B)** Disruption of *atfs-1* decreased resistance to chronic oxidative stress (4 mM paraquat). *atfs-1(et17)* mutants showed increased resistance to chronic oxidative stress, whereas *atfs-1(et15)* mutants had decreased resistance. **(C)** Resistance to heat stress (37°C) was not enhanced by activation of ATFS-1, whereas disruption of *atfs-1* decreased heat stress resistance. **(D)** Constitutive activation of ATFS-1 increased resistance to ER stress (50 μM tunicamycin), whereas disruption of *atfs-1* had no effect. **(E)** Activation of ATFS-1 increased resistance to osmotic stress (500 mM NaCl), whereas disruption of *atfs-1* decreased osmotic stress resistance. **(F)** Constitutively active *atfs-1* mutants show increased resistance to anoxia (75 h), whereas *atfs-1* deletion mutants exhibit a trend towards decreased anoxia resistance. **(G)** Activation of ATFS-1 increased resistance to *Pseudomonas aeruginosa* toxin in a fast kill assay. A slow kill assay in which worms die from internal accumulation of *P. aeruginosa* was performed according to two established protocols. **(H)** At 25°C, *atfs-1(et17)* mutants showed a small decrease in resistance to bacterial pathogens (PA14), wheras *atfs-1(gk3094)* mutants showed a small increase in resistance. **(I)** At 20°C, both *atfs-1(et17)* and *atfs-1(gk3094)* mutants exhibited a small increase in resistance to bacterial pathogens. Data for WT and *atfs-1(gk3094)* in panel (I) are from Campos et al (2021) as these strains were used as controls for two separate experiments that were performed at the same time. Data information: Error bars indicate SEM. *$P < 0.05$, **$P < 0.01$, ***$P < 0.001$. Statistical analysis for panels (A, B, D, H, I) were performed using the log-rank test. Statistical analysis for panels C and G were performed using a two-way ANOVA with Bonferroni post hoc test. Statistical analysis for panels (E, F) was performed using a one-way ANOVA with Bonferroni post hoc test. The number of replicates, N, and statistical analysis can be found in Table S6.

All together, these data indicate that activation of ATFS-1 can protect against oxidative stress, ER stress, osmotic stress, anoxia, and bacterial pathogens but not heat stress. They also show that ATFS-1 is required for resistance to oxidative stress, heat stress, osmotic stress, and anoxia in wild-type worms.

### Long-lived genetic mutants up-regulate ATFS-1 target genes

We previously showed that ATFS-1 target genes are up-regulated in three long-lived mitochondrial mutants: *clk-1, isp-1,* and *nuo-6* (Lakowski & Hekimi, 1996; Feng et al, 2001; Yang & Hekimi, 2010b; Wu et al, 2018). To determine if ATFS-1 target genes are specifically up-regulated in long-lived mitochondrial mutants, or if they are also up-regulated in other long-lived mutants, we compared genes up-regulated by ATFS-1 activation with gene expression in six additional long-lived mutants, which act through other longevity-promoting

pathways. These long-lived mutants included *sod-2* mutants, which act through increasing mitochondrial ROS (Van Raamsdonk & Hekimi, 2009); *daf-2* mutants, which have reduced insulin/IGF1 signaling (Kenyon et al, 1993); *glp-1* mutants, which have germ line ablation (Hsin & Kenyon, 1999); *ife-2* mutants, which have reduced translation (Hansen et al, 2007); *osm-5* mutants, which have reduced chemosensation (Apfeld & Kenyon, 1999); and *eat-2* mutants, which have dietary restriction (Lakowski & Hekimi, 1998).

After identifying differentially expressed genes in each of these long-lived mutants, we compared the differentially expressed genes with genes up-regulated by ATFS-1 activation. We defined ATFS-1-up-regulated genes in two ways: (1) genes that are up-regulated by *spg-7* RNAi in an ATFS-1–dependent manner (Nargund et al, 2012) and (2) genes that are up-regulated in a constitutively active *atfs-1* mutant (*et15;* [Wu et al, 2018]).

The majority of the long-lived mutants examined had a significant enrichment of ATFS-1 target genes. Genes up-regulated by *spg-7* RNAi in an ATFS-1–dependent manner were significantly enriched in *clk-1* mutants (6.7-fold enrichment), *isp-1* mutants (6.0-fold enrichment), *sod-2* mutants (5.5-fold enrichment), *nuo-6* mutants (4.1-fold enrichment), *daf-2* mutants (2.6-fold enrichment), *glp-1* mutants (2.0-fold enrichment), and *ife-2* mutants (1.5-fold enrichment) (Fig 5). We did not find a significant enrichment of *spg-7* RNAi-induced ATFS-1 targets in *osm-5* and *eat-2* worms (Fig 5). Similarly, genes up-regulated in the constitutively active *atfs-1(et15)* mutant were significantly enriched in *isp-1* mutants (3.5-fold enrichment), *sod-2* mutants (3.4-fold enrichment), *clk-1* mutants (3.3-fold enrichment), *nuo-6* mutants (2.5-fold enrichment), *daf-2* mutants (2.4-fold enrichment), *glp-1* mutants (1.8-fold enrichment), *ife-2* mutants (1.8-fold enrichment), and *eat-2* mutants (1.5-fold enrichment) (Fig S9). We did not observe a significant enrichment of ATFS-1 target genes in *osm-5* mutants (Fig S9).

Overall, these results indicate that ATFS-1 target genes are up-regulated in multiple long-lived mutants, including mutants in which mitochondrial function is not directly disrupted. Interestingly, in six of the seven strains exhibiting a significant enrichment of ATFS-1–modulated genes (all except *ife-2*, where the role of ROS has not been tested), there is an increase in ROS that contributes to their longevity as treatment with antioxidants decreases their lifespan (Van Raamsdonk & Hekimi, 2009; Yang & Hekimi, 2010a; Zarse et al, 2012; Wei & Kenyon, 2016). This observation is consistent with the idea that ROS/oxidative stress is sufficient to activate the mitoUPR. As we have previously shown that exposure to a mild heat stress (35°C, 2 h) or osmotic stress (300 mM, 24 h) can extend lifespan but does not increase expression of the ATFS-1 target gene *hsp-6* (Dues et al, 2016), it appears that only specific genes or interventions that extend longevity result in the up-regulation of ATFS-1 target genes.

## Constitutively active *atfs-1* mutants have decreased lifespan despite enhanced resistance to stress

Having shown that ATFS-1 target genes are activated in multiple long-lived mutants, we sought to determine if ATFS-1 activation is sufficient to increase lifespan and whether the presence of ATFS-1 is required for normal longevity in wild-type worms. Despite having increased resistance to multiple stressors, both constitutively active *atfs-1* mutants (*et15* and *et17*) have decreased lifespan compared with wild-type worms (Fig 6A and B), which is consistent with a previous study finding shortened lifespan in *atfs-1(et17)* and *atfs-1(et18)* worms (Bennett et al, 2014). Despite having decreased resistance to multiple stressors, *atfs-1* deletion mutants (*gk3094*) had a lifespan comparable with wild-type worms (Fig 6C), as we previously observed (Wu et al, 2018). Combined, this indicates that constitutive activation of ATFS-1 does not increase lifespan in a wild-type background, despite having an important role in stress resistance.

## Discussion

Mitochondria are vital for organismal health as they perform multiple crucial functions within the cell including energy generation, metabolic

**Figure 5.  Multiple long-lived mutants from different pathways of lifespan extension show up-regulation of ATFS-1–dependent genes.**
To determine the extent to which long-lived genetic mutants from different pathways of lifespan extension show differential expression of ATFS-1 target genes, we compared genes that are up-regulated in nine different long-lived mutants to a published list of *spg-7* RNAi-up-regulated, ATFS-1–dependent target genes (Nargund et al, 2012). *clk-1*, *isp-1*, *nuo-6*, *sod-2*, *daf-2*, *glp-1*, and *ife-2* worms all show a highly significant degree of overlap with genes up-regulated by *spg-7* RNAi in an ATFS-1–dependent manner. The grey circles represent the 366 genes that are up-regulated by *spg-7* RNAi in an ATFS-1–dependent manner. Turquoise circles are genes that are significantly up-regulated in the indicated long-lived mutant based upon our RNA sequencing data. The number of unique and overlapping genes is indicated. Percent overlap is calculated as the number of genes in common between the two gene sets divided by the total number of genes that are up-regulated by *spg-7* RNAi in an ATFS-1–dependent manner. Enrichment is calculated as the number of overlapping genes observed divided by the number of overlapping genes predicted if genes were chosen randomly. *P*-values indicate the significance of the difference between the observed number of overlapping genes between the two gene sets, and the expected number of overlapping genes if the genes were picked at random.

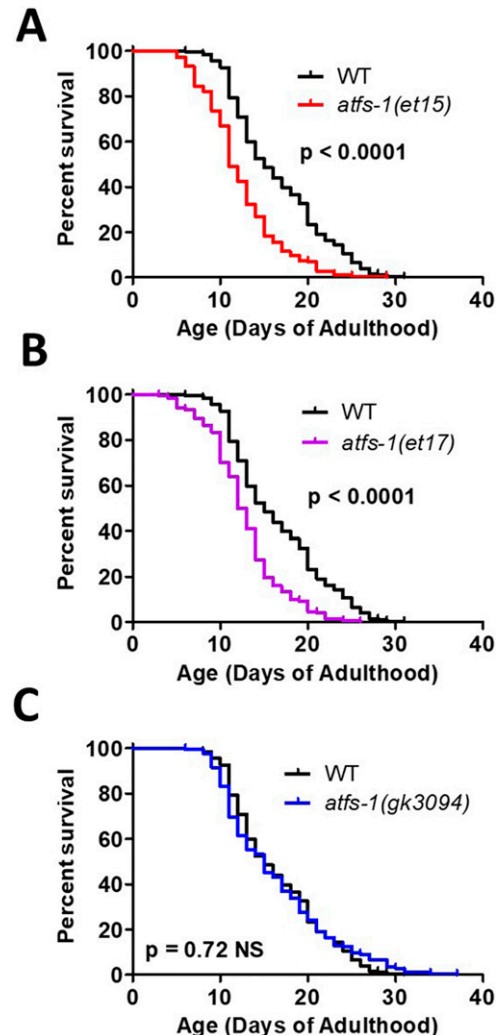

**Figure 6. Activation of ATFS-1 does not increase lifespan.**
To determine the effect of ATFS-1 on aging, we quantified the lifespan of an *atfs-1* deletion mutant and two constitutively active *atfs-1* mutants. **(A, B)** Both constitutively active *atfs-1* mutants, *et15* and *et17*, have a significantly decreased lifespan compared with wild-type worms. **(C)** Disruption of *atfs-1* does not affect lifespan compared with wild-type worms. *atfs-1(gk3094)* is a loss of function mutant resulting from a deletion. *atfs-1(et15)* and *atfs-1(et17)* are constitutively active gain-of-function mutants. Data information: Statistical analysis was performed using the log-rank test. Statistical analysis, number of replicates, N, and raw lifespan data are available in Table S6.

reactions, and intracellular signaling. Therefore, maintenance of mitochondrial function during times of acute stress and throughout normal aging is important for cell and organismal survival. The mitoUPR is a conserved pathway that facilitates restoration of mitochondrial homeostasis after internal or external stressors. In this work, we demonstrate a crucial role for the mitoUPR transcription factor ATFS-1 in the genetic response to external stressors, which ultimately promotes survival of the organism.

Throughout these studies, we utilized two different constitutively active *atfs-1* mutants—*et15* and *et17*. These two mutants contain point mutations in the MTS and differ only by one two amino acids (*et15*: G6E, *et17*: R4H) (Rauthan et al, 2013). Although *atfs-1(et15)* and

*atfs-1(et17)* mutants generally behave similarly, they do exhibit differences, most notably in resistance to chronic oxidative stress and resistance to bacterial pathogens in the slow kill assay. These differences may result from *atfs-1(et15)* mutants having more extensive changes in gene expression than *atfs-1(et17)* mutants (6,227 differentially expressed genes versus 958 differentially expressed genes) (Wu et al, 2018). The *et15* mutation may be more disruptive to the MTS than *et17*, thereby resulting in increased nuclear localization and more widespread changes in gene expression.

### ATFS-1 is not required for normal longevity

A number of studies have directly or indirectly examined the role of the mitoUPR and ATFS-1 in longevity. In these studies, activation of the mitoUPR was typically measured using a mitoUPR reporter strain expressing GFP under the promoter of *hsp-6*, which is a target gene of ATFS-1 and the mitoUPR.

A relationship between the mitoUPR and longevity was first supported by the observation that disruption of the mitochondrial electron transport chain due to RNAi knockdown of the cytochrome c oxidase-1 (*cco-1*) gene resulted in both activation of the mitoUPR (Yoneda et al, 2004; Durieux et al, 2011) and increased lifespan (Dillin et al, 2002). Since then, other lifespan-extending mutations have also been shown to activate the mitoUPR, including three long-lived mitochondrial mutants, *clk-1*, *isp-1*, and *nuo-6* (Wu et al, 2018).

To explore this relationship in a more comprehensive manner, Runkel et al (2014) compiled a list of genes that activate the mitoUPR and examined their effect on lifespan. Of the 99 genes reported to activate the mitoUPR, 58 genes result in increased lifespan, although only 7 result in decreased lifespan (Runkel et al, 2014). Bennet et al (2014) performed an RNAi screen to identify RNAi clones that increase expression of a mitoUPR reporter strain (*hsp-6p:: GFP*) and quantified the effect of a selection of the mitoUPR-inducing clones on lifespan (Bennett et al, 2014). Of the 19 examined RNAi clones, 10 RNAi clones increased lifespan, while 6 decreased lifespan (Bennett et al, 2014). Using a similar approach to screen for compounds that activate a mitoUPR reporter strain (*hsp-6p::GFP*), metolazone was identified as a compound that activates the mitoUPR and extends lifespan in an ATFS-1–dependent manner (Ito et al, 2021). Combined, these results indicate that there are multiple genes or interventions which activate the mitoUPR and extend longevity, but there are also instances where these phenotypes are uncoupled.

Multiple experiments including the present study have examined the effect of the mitoUPR on lifespan directly by either increasing or decreasing the expression of components of the mitoUPR. RNAi knockdown of *atfs-1* expression does not decrease wild-type lifespan (Bennett et al, 2014; Tian et al, 2016; Wu et al, 2018) nor do deletions in the *atfs-1* gene decrease wild-type lifespan (Fig 6; [Bennett et al, 2014; Wu et al, 2018]). Thus, despite mitoUPR activation being correlated with longevity, ATFS-1 is not required for normal lifespan in a wild-type animal.

### ATFS-1 mediates lifespan extension in long-lived mutants

Although ATFS-1 is dispensable for wild-type lifespan, it is required for lifespan extension of multiple long-lived mutants. Longevity can be extended by disrupting mito-nuclear protein balance through

knocking down the expression of mitochondrial ribosomal protein S5 (*mrsp-5*), which also increases the expression of the mitoUPR target gene *hsp-6*. The magnitude of the lifespan extension caused by *mrsp-5* RNAi is decreased by knocking down the key mitoUPR component gene *haf-1* or *ubl-5* (Houtkooper et al, 2013). In the long-lived mitochondrial mutant *nuo-6*, disruption of *atfs-1* completely reverts the long lifespan to wild-type length, and treatment with *atfs-1* RNAi has similar effects (Wu et al, 2018). In the mitochondrial mutant *isp-1*, knocking down a key initiator of mitoUPR, *ubl-5*, decreases their long lifespan but has no effect on the lifespan of wild-type worms (Durieux et al, 2011). In contrast, it has been reported that knockdown of *atfs-1* using RNAi does not decrease *isp-1* lifespan (Bennett et al, 2014). However, it is possible that in the latter study, the magnitude of knockdown may not have been sufficient to have effects on lifespan as lifelong exposure to *atfs-1* RNAi prevents larval development of *isp-1* worms (Baker et al, 2012; Wu et al, 2018). Similarly, differing results have been obtained for the requirement of the mitoUPR in the extended lifespan resulting from *cco-1* knockdown. Although it has been reported that mutation of *atfs-1* does not decrease the lifespan of worms treated with *cco-1* RNAi, despite preventing activation of mitoUPR reporter (Bennett et al, 2014), a subsequent study found that *atfs-1* RNAi decreases the extent of lifespan extension resulting from *cco-1* RNAi (Tian et al, 2016). Although differing results have been observed in some cases, overall, these studies suggest that ATFS-1 and the mitoUPR have a role in mediating the lifespan extension in a subset of long-lived mutants.

Despite the fact that long-lived mutants with chronic activation of the mitoUPR depend on ATFS-1 for their long lifespan, our current results using the constitutively active *atfs-1(et15)* and *atfs-1(et17)* mutants, as well as previous results using constitutively active *atfs-1* mutants (*et17* and *et18*), show that constitutive activation of ATFS-1 in wild-type worms results in decreased lifespan (Fig 6) (Bennett et al, 2014). This may be partially due to activation of ATFS-1, increasing the proportion of damaged mtDNA when heteroplasmy exists (Lin et al, 2016). Consistent with this finding, overexpression of the mitoUPR target gene *hsp-60* also leads to a small decrease in lifespan (Jeong et al, 2017). In contrast, overexpression of a different mitoUPR target gene, *hsp-6*, is sufficient to increase lifespan (Yokoyama et al, 2002). It has also been shown that a hypomorphic reduction-of-function mutation allele of *hsp-6* (*mg583*) also increases lifespan, whereas *hsp-6* null mutations are thought to be lethal (Mao et al, 2019). Combined, these results indicate that chronic activation of the mitoUPR is mildly detrimental for wild-type lifespan, but that modulation of specific target genes can be beneficial.

It is important to note that the lifespan assays completed in this study and previous studies were completed under laboratory conditions, which are believed to be relatively unstressed. It is possible that constitutive activation of ATFS-1 may increase lifespan in an uncontrolled environment where worms encounter external stressors, as observed with our various stress assays. The magnitude of ATFS-1 activation may impact its effect on stress resistance and lifespan. Perhaps, a milder activation of ATFS-1 will be more beneficial with respect to lifespan, which could be determined through dose–response experiments involving RNAi-mediated knockdown of *atfs-1* in the constitutively active *atfs-1* mutants.

## ATFS-1 is necessary for stress resistance in wild-type animals

Although ATFS-1 is not required for longevity in wild-type animals, it plays a significant role in protecting animals against exogenous stressors. Disrupting *atfs-1* function decreases organismal resistance to oxidative stress, heat stress, osmotic stress, and anoxia (Fig 4). Additionally, we previously determined that inhibiting *atfs-1* in long-lived *nuo-6* worms completely suppressed the increased resistance to oxidative stress, osmotic stress, and heat stress typically observed in that mutant (Wu et al, 2018) and that disruption of *atfs-1* in Parkinson's disease mutants *pdr-1* and *pink-1* decreased their resistance to oxidative stress, osmotic stress, heat stress, and anoxia (Cooper et al, 2017). Combined, these results demonstrate that ATFS-1 is required for resistance to multiple exogenous stressors.

Even though ATFS-1 is required for the up-regulation of stress response genes in response to bacterial pathogens (Fig 3), disruption of *atfs-1* (*gk3094* mutation) did not decrease bacterial pathogen resistance. Similarly, another *atfs-1* deletion mutation (*tm4919*) was found not to affect survival during exposure to *P. aeruginosa* (Pellegrino et al, 2014). In contrast, Jeong et al (2017) did observe decreased bacterial pathogen survival in *atfs-1(gk3094)* mutants (Jeong et al, 2017). Knocking down *atfs-1* through RNAi also inconsistently decreased survival on *P. aeruginosa* (e.g., Fig 3A versus Fig 3H in Pellegrino et al [2014]). It is unclear why disruption of *atfs-1* has a variable effect on bacterial pathogen resistance but may result from subtle differences in the way the assay is conducted.

Consistent with our finding that *atfs-1* deletion does not decrease resistance to bacterial pathogens in wild-type worms, we have shown that baseline expression of innate immunity genes in wild-type animals is also not affected by disruption of *atfs-1* (Campos et al, 2021). In contrast, disrupting genes involved in the p38-mediated innate immune signaling pathway does decrease resistance to bacterial pathogens and does decrease the expression of innate immunity genes in a wild-type background (Campos et al, 2021). Combined, this indicates that baseline levels of innate immunity gene expression and bacterial pathogen resistance are dependent on the p38-mediated innate immune signaling pathway and are not dependent on ATFS-1. In contrast, the expression of innate immunity genes can be enhanced by activation of ATFS-1, either in *nuo-6* mutants (Campos et al, 2021) or constitutively active *atfs-1* mutants (Fig 4G).

Decreasing the expression of a downstream ATFS-1 target gene, *hsp-60*, by RNAi caused a robust decrease in organismal survival on *P. aeruginosa* (Jeong et al, 2017). As we have previously found that disrupting *atfs-1* induces up-regulation of other protective cellular pathways (Wu et al, 2018), and others have observed a similar phenomenon when a mitoUPR downstream target, *hsp-6*, is disrupted (Kim et al, 2016), it is possible that the up-regulation of other stress pathways may compensate for the inhibition of the mitoUPR in *atfs-1* deletion mutants, ultimately yielding wild-type or increased levels of resistance to bacterial pathogens and hiding the normal role of the mitoUPR in resistance to bacterial pathogens.

### Activation of ATFS-1 enhances resistance to exogenous stressors

In this work, we show that constitutive activation of ATFS-1 (*atfs-1(et15)* and *atfs-1(et17)* mutants) is sufficient to increase resistance to multiple different exogenous stressors, including oxidative stress, ER stress, osmotic stress, anoxia, and bacterial pathogens. Previous studies have shown that activating the mitoUPR, either through *spg-7* RNAi or through a constitutively active *atfs-1(et15)* mutant, decreased risk of death after anoxia–reperfusion (Pena et al, 2016) and that constitutively active *atfs-1(et18)* mutants have increased resistance to *P. aeruginosa* (Pellegrino et al, 2014). Overexpression of the mitoUPR target gene *hsp-60* also increases resistance to *P. aeruginosa* (Jeong et al, 2017). These results support a clear role for ATFS-1 in surviving external stressors.

Although ATFS-1 activation protects against multiple external stressors, not all of these stressors activate ATFS-1. Previously, we exposed a mitoUPR reporter strain (*hsp-6p::GFP*) to heat stress, cold stress, osmotic stress, anoxia, oxidative stress, starvation, ER stress, and bacterial pathogens, and only oxidative stress increased mitoUPR activity (Dues et al, 2016). As the constitutively active *atfs-1* mutants (*et15* and *et17*) exhibit activation of the mitoUPR under unstressed conditions (e.g., up-regulation of *hsp-6* in Fig 1A; up-regulation of many other stress pathway target genes Fig 2; increased fluorescence of *hsp-6* and *hsp-60* reporter strains in Rauthan et al [2013]), it is likely that the activation of the mitoUPR and downstream stress response pathways under unstressed conditions is primarily responsible for the increased resistance to stress that we observe in the constitutively active *atfs-1* mutants.

### ATFS-1 up-regulates target genes of multiple stress response pathways

In exploring the mechanism by which ATFS-1 and the mitoUPR modulate stress resistance, we found that activation of ATFS-1, through mild impairment of mitochondrial function (*nuo-6*) or through constitutive activation of ATFS-1 (*atfs-1(et15)*), causes up-regulation of genes involved in multiple stress response pathways, including the ER-UPR pathway, the Cyto-UPR pathway, the DAF-16–mediated stress response pathway, the SKN-1–mediated oxidative stress response pathway, the HIF-mediated hypoxia response pathway, the p38-mediated innate immune response pathway, and antioxidant genes (Fig 2). These findings are consistent with those of earlier work demonstrating a role for ATFS-1 in up-regulating innate immunity genes. Pellegrino et al (2014) reported a 16% (59/365 genes) overlap between genes up-regulated by activation of the mitoUPR through treatment with *spg-7* RNAi and genes up-regulated by exposure a bacterial pathogen (Pellegrino et al, 2014). A connection between the mitoUPR and the innate immunity pathway was also suggested by the finding that overexpression of a mitoUPR downstream target, *hsp-60*, increases expression of three innate immunity genes: *T24B8.5/sysm-1*, *C17H12.8*, and *K08D8.5* (Jeong et al, 2017). Our results clearly indicate that the role of ATFS-1 in stress response pathways is not limited to the innate immunity but extends to multiple stress response pathways, thereby providing a mechanistic basis for the effect of ATFS-1 on resistance to stress.

Although our results do not definitively distinguish between direct or indirect regulation of genes and other stress response pathways by ATFS-1, analysis of previous CHiP-seq experiments demonstrates that ATFS-1 can bind to the same genes as other stress-responsive transcription factors including HSF-1, DAF-16, HIF-1, SKN-1, and ATF-7. The ability of ATFS-1 to bind to these genes suggests that ATFS-1 may be able to directly regulate a subset of target genes of other stress response pathways.

### Conclusions

The mitoUPR is required for animals to survive exposure to exogenous stressors, and activation of this pathway is sufficient to enhance resistance to stress (Table S5). In addition to up-regulating genes involved in restoring mitochondrial homeostasis, the mitoUPR increases stress resistance by up-regulating the target genes of multiple stress response pathways. Although increased stress resistance has been associated with long lifespan, and multiple long-lived mutants exhibit activation of the mitoUPR, constitutive activation of ATFS-1 shortens lifespan while increasing resistance to stress, indicating that the role of ATFS-1 in stress resistance can be experimentally dissociated from its role in longevity. Overall, this work highlights the importance of the mitoUPR in not only protecting organisms from internal stressors but also improving organismal survival upon exposure to external stressors.

# Materials and Methods

### Strains

*C. elegans* strains were obtained from the *Caenorhabditis* Genetics Center (CGC): N2 (wild-type), *nuo-6(qm200)*, *atfs-1(gk3094)*, *nuo-6(qm200);atfs-1(gk3094)*, *atfs-1(et15)*, *atfs-1(et17)*, *ife-2 (ok306)*, *clk-1(qm30)*, *sod-2(ok1030)*, *eat-2(ad1116)*, *osm-5(p813)*, *isp-1(qm150)*, *daf-2(e1370)*, and *glp-1(e2141)*. Strains were maintained at 20°C on nematode growth medium (NGM) plates seeded with OP50 *Escherichia coli*. *atfs-1(et15)* and *atfs-1(et17)* were outcrossed 10 times (Rauthan et al, 2013), and *atfs-1(gk3094)* were outcrossed six times. Young adult worms are picked on day 1 of adulthood before egg laying begins. The worms were not synchronized but picked visually as close to the L4-adult transition as possible.

### Gene expression in response to stress

#### *Stress treatment*
Young adult worms were subject to different stress before mRNA was collected. For heat stress, worms were incubated at 35°C for 2 h and 20°C for 4 h. For oxidative stress, worms were transferred to plates containing 4 mM paraquat and 100 0$\mu$M FUdR for 48 h. FUdR was used for these samples because (1) with the 2-d duration of this stress, worms can produce progeny which would complicate the collection of the experimental worms; and (2) 4 mM paraquat often results in internal hatching of progeny when FUdR is absent, which might have affected the results. Because FUdR has the potential to

alter gene expression, the control worms for the 48-h 4 mm paraquat stress were also treated with 100 μM FUdR. For ER stress, worms were transferred to plates containing 5 μg/ml tunicamycin for 24 h. For osmotic stress, worms were transferred to plates containing 300 mM NaCl and left for 24 h. For bacterial pathogen stress, worms were transferred to plates seeded with *P. aeruginosa* strain PA14 and left for 4 h. For anoxic stress, worms were put in BD Bio-Bag Type A Environmental Chambers (Becton, Dickinson and Company) for 24 h and left to recover for 4 h. For unstressed control conditions, worms were collected at the young adult stage for heat stress and bacterial pathogens; 24 h after the young adult stage for osmotic stress, ER stress and anoxia; and 48 h after the young adult stage for oxidative stress.

### RNA isolation
RNA was harvested, as described previously (Schaar et al, 2015). The plates of worms were washed three times using M9 buffer to remove bacteria and resuspended in TRIzol reagent. Worms were frozen in a dry ice/methanol bath and then thawed three times and left at room temperature for 15 min. Chloroform was added to the tubes, and the mixture was left to sit at room temperature for 3 min. The tubes were then centrifuged at 12,000*g* for 15 min at 4°C. The upper phase containing the RNA was transferred to a new tube, mixed with isopropanol, and allowed to sit at room temperature for 10 min. The tubes were centrifuged at 12,000*g* for 10 min at 4°C. The RNA pellet was washed with 75% ethanol and resuspended in RNAse-free water.

### Quantitative RT-PCR
mRNA was converted to cDNA using a High-Capacity cDNA Reverse Transcription kit (Life Technologies/Invitrogen), as described previously (Machiela et al, 2016). qPCR was performed using a PowerUp SYBR Green Master Mix kit (Applied Biosystems) in a Viia 7 RT-PCR machine from Applied Biosystems. All experiments were performed with at least three biological replicates collected from different days. mRNA levels were normalized to *act-3* levels and then expressed as percentage of wild-type. Primer sequences are as follows:

*gst-4* (CTGAAGCCAACGACTCCATT, GCGTAAGCTTCTTCCTCTGC),
*hsp-4* (CTCGTGGAATCAACCCTGAC, GACTATCGGCAGCGGTAGAG),
*hsp-6* (CGCTGGAGATAAGATCATCG, TTCACGAAGTCTCTGCATGG),
*hsp-16.2* (CCATCTGAGTCTTCTGAGATTGTT, CTTTCTTTGGCGCTTCAATC),
*sod-3* (TACTGCTCGCACTGCTTCAA, CATAGTCTGGGCGGACATTT),
*sod-5* (TTCCACAGGACGTTGTTTCC, ACCATGGAACGTCCGATAAC),
*nhr-57* (GACTCTGTGTGGAGTGATGGAGAG, GTGGCTCTTGGTGTCAATTTCGGG),
*gcs-1* (CCACCAGATGCTCCAGAAAT, TGCATTTTCAAAGTCGGTC),
*trx-2* (GTTGATTTCCACGCAGAATG, TGGCGAGAAGAACACTTCCT),
*Y9C9A.8* (CGGGGATATAACTGATAGAATGG, CAAACTCTCCAGCTTCCAACA),
*T24B8.5* (TACACTGCTTCAGAGTCGTG, CGACAACCACTTCTAACATCTG),
*clec-67* (TTTGGCAGTCTACGCTCGTT, CTCCTGGTGTGTCCCATTTT),
*dod-22* (TCCAGGATACAGAATACGTACAAGA, GCCGTTGATAGTTTCGGTGT),
*ckb-2* (GCATTTATCCGAGACAGCGA, GCTTGCACGTCCAAATCAAC),
*act-3* (TGCGACATTGATATCCGTAAGG, GGTGGTTCCTCCGGAAAGAA).

## RNA sequencing and bioinformatics analysis

RNA sequencing was performed previously (Dues et al, 2017; Senchuk et al, 2018), and raw data are available on the National

Center for Biotechnology Information (NCBI) Gene Expression Omnibus (GEO): GSE93724 (Senchuk et al, 2018), GSE110984 (Wu et al, 2018). Bioinformatics analysis for this study was used to determine differentially expressed genes and identify the degree and significance of overlaps between genes sets.

### Determining differentially expressed genes
Samples were processed using an RNA-seq pipeline based on the bcbio-nextgen project (https://bcbio-nextgen.readthedocs.org/en/latest/). We examined raw reads for quality issues using FastQC (http://www.bioinformatics.babraham.ac.uk/projects/fastqc/) in order to ensure library generation, and sequencing data were suitable for further analysis. If necessary, we used cutadapt https://cutadapt.readthedocs.io/en/stable/ to trim adapter sequences, contaminant sequences such as polyA tails, and low-quality sequences from reads. We aligned trimmed reads to the Ensembl build WBcel235 (release 90) of the *C. elegans* genome using STAR (Dobin et al, 2013). We assessed the quality of alignments by checking for evenness of coverage, ribosomal RNA content, genomic context of alignments (e.g., alignments in known transcripts and introns), complexity, and other quality checks. To quantify expression, we used Salmon (Patro et al, 2017) to find transcript-level abundance estimates and then collapsed down to the gene level using the R Bioconductor package tximport (Soneson et al, 2015). Principal components analysis and hierarchical clustering methods were used to validate clustering of samples from the same batches and across different mutants. We used the R Bioconductor package DESeq2 (Love et al, 2014) to find differential expression at the gene level. For each wild-type mutant comparison, we identified significant genes with an false discovery rate threshold of 0.01. Lastly, we included batch as a covariate in the linear model for datasets in which experiments were run across two batches.

### Venn diagrams
Weighted Venn diagrams were produced by inputting gene lists into BioVenn (https://www.biovenn.nl/). Percentage overlap was determined by dividing the number of genes in common between the two gene sets by the gene list with the smaller gene list.

### Significance of overlap and enrichment
The significance of overlap between two gene sets was determined by comparing the actual number of overlapping genes with the expected number of overlapping genes based on the sizes of the two gene sets (expected number = number of genes in set 1 × number of genes in set 2/number of genes in genome detected). Enrichment was calculated as the observed number of overlapping genes/the expected number of overlapping genes if genes were chosen randomly.

## Resistance to stress

For acute oxidative stress, young adult worms were transferred onto plates with 300 μM juglone and survival was measured every 2 h for a total of 10 h. For chronic oxidative stress, young adult worms were transferred onto plates with 4 mM paraquat and 100 μM FUdR and survival was measured daily until death.

For heat stress, young adult worms were incubated in 37°C and survival was measured every 2 h for a total of 10 h. For osmotic stress, young adult worms were transferred to plates containing 450 or 500 mM NaCl and survival was measured after 48 h. For anoxic stress, plates with young adult worms were put into BD Bio-Bag Type A Environmental Chambers for 75 h and survival was measured after a 24-h recovery period.

Resistance to ER stress was tested by transferring young adult worms to agar plates containing either 50 μg/ml tunicamycin (654380; EMD Millipore) in 0.5% DMSO (472301; Sigma-Aldrich) or 0.5% DMSO only at 20°C. Survival was measured every day until death.

Two different bacterial pathogenesis assays involving *P. aeruginosa* strain PA14 were performed. In the slow kill assay, worms are thought to die from intestinal colonization of the pathogenic bacteria, whereas in the fast kill assay, worms are thought to die from a toxin secreted from the bacteria (Kirienko et al, 2014). The slow kill assay was performed, as described previously (Pellegrino et al, 2014; Wu et al, 2019). In the first protocol (Pellegrino et al, 2014), PA14 cultures were grown overnight and seeded to center of a 35-mm NGM agar plate. The plates were left to dry overnight and then incubated in 37°C for 24 h. The plates were left to adjust to room temperature before ~40 L4 worms were transferred onto the plates. The assay was conducted 25°C, and the plates were checked twice a day until death. In the second protocol (Wu et al, 2019), overnight PA14 culture were seeded to the center of a 35-mm NGM agar plate containing 20 mg/l FUdR. The plates were incubated at 37°C overnight and then at room temperature overnight before ~40 d, three adults were transferred onto these plates. The assay was conducted 20°C, and the plates were checked daily until death. The fast kill pathogenesis assay was performed, as described previously (Kirienko et al, 2014). PA14 cultures were grown overnight and seeded to peptone–glucose–sorbitol agar plates. Seeded plates were left to dry for 20 min at room temperature before incubation at 37°C for 24 h and then at 23°C for another 24 h. Approximately 30 L4 worms were transferred onto the plates and were scored as dead or alive at 2, 4, 6, 8, and 24 h. Fast kill plates were kept at 23°C in between scoring time points.

### Lifespan

All lifespan assays were performed at 20°C. Lifespan assays included FUdR to limit the development of progeny and the occurrence of internal hatching. Based on our previous studies, a low concentration of FUdR (25 mM) was used to minimize potential effects of FUdR on lifespan (Van Raamsdonk & Hekimi, 2011). Animals were excluded from the experiment if they crawled off the plate or died of internal hatching of progeny or expulsion of internal organs.

### Statistical analysis

All of our statistical analyses are provided in Table S6 including the number of replicates, worms per replicate, statistical test utilized, and all *P*-values. To ensure unbiased results, all experiments were conducted with the experimenter blinded to the genotype of the worms. For all assays, a minimum of three biological replicates of randomly selected worms from independent populations of worms on different days were used. For analysis of lifespan, oxidative stress, and bacterial pathogen stress, a log-rank test was used. For analysis of heat stress, repeated measures ANOVA was used. For analysis of osmotic stress and anoxic stress, a one-way ANOVA with Dunnett's multiple comparisons test was used. For quantitative PCR results, we used a two-way ANOVA with the Bonferroni post hoc test. For all bar graphs, error bars indicate the standard error of the mean and bars indicate the mean.

## Data Availability

RNA-seq data have been deposited on GEO: GSE93724, GSE110984. All other data and strains generated in the current study are included with the article or available from the corresponding author on request.

## Supplementary Information

## Acknowledgements

Some strains were provided by the CGC, which is funded by the National Institutes of Health (NIH) Office of Research Infrastructure Programs (P30 OD010440). We would also like to acknowledge the *C. elegans* knockout consortium and the National Bioresource Project of Japan for providing strains used in this research. This work was supported by the Canadian Institutes of Health Research (CIHR; http://www.cihr-irsc.gc.ca/; JM Van Raamsdonk), the Natural Sciences and Engineering Research Council of Canada (NSERC; https://www.nserc-crsng.gc.ca/index_eng.asp; JM Van Raamsdonk), and the National Institute of General Medical Sciences (NIGMS; https://www.nigms.nih.gov/; JM Van Raamsdonk) by grant number R01 GM121756. JM Van Raamsdonk is the recipient of a Senior Research Scholar career award from the Fonds de Recherche du Québec Santé (FRQS) and Parkinson Quebec. A Traa received scholarships from the NSERC and FRQS. SK Soo received a scholarship from the FRQS. PD Rudich received a fellowship award from the FRQS. The funders had no role in study design, data collection and analysis, decision to publish, or preparation of the manuscript.

### Author Contributions

SK Soo: formal analysis, investigation, visualization, methodology, and writing—original draft, review, and editing.
A Traa: formal analysis, investigation, visualization, methodology, and writing—review and editing.
PD Rudich: formal analysis, investigation, visualization, methodology, and writing—review and editing.
M Mistry: formal analysis, investigation, visualization, and methodology.
JM Van Raamsdonk: conceptualization, formal analysis, supervision, funding acquisition, investigation, visualization, methodology, project administration, and writing—original draft, review, and editing.

## Conflict of Interest Statement

The authors declare that they have no conflict of interest.

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
