## [Reviewer comments · Life Science Alliance]

Life Science Alliance

Activation of mitochondrial unfolded protein response protects against multiple exogenous stressors

Sonja Soo, Annika Traa, Paige Rudich, Meeta Mistry, and Jeremy Van Raamsdonk
DOI: <https://doi.org/10.26508/lsa.202101182>

Corresponding author(s): Jeremy Van Raamsdonk, McGill University

Review Timeline:	Submission Date:	2021-07-30
	Editorial Decision:	2021-08-18
	Revision Received:	2021-09-10
	Accepted:	2021-09-20

Transaction Report:

Please note that the manuscript was reviewed at *Review Commons* and these reports were taken into account in the decision-making process at *Life Science Alliance*.

Review
COMMONS

Reviewer #1:****Major concerns:****

1) This manuscript has some overlap with another manuscript from the same group recently submitted to EMBO Reports. Although I believe both manuscripts have sufficient elements to justify publication of two papers, I strongly recommend that these publications are made back-to-back and they should be discussed in context with one-another.

We agree that this manuscript is distinct from but highly complementary to our manuscript on innate immunity in the long-lived mitochondrial mutants, which has been invited for revision at *EMBO Reports*. According to this suggestion, we have arranged for these papers to be considered for publication at the same time in *EMBO Reports* and *Life Science Alliance*. We have updated the discussions of both manuscripts to incorporate the findings of the other manuscript.

2) How is ATFS-1 function regulated in long-lived worms or under multiple stress conditions? Is there a common regulator such as oxidative stress or mitochondrial dysfunction? Both manuscripts would benefit from a clear understanding on how ATFS-1 is controlled under conditions where mitochondrial function is altered. Is mitoUPR required for this activation? If so, is mitoUPR upregulated in all interventions where ATFS-1 has been shown to play a role in stress response.

We have previously used a reporter strain to determine which external stressors activate ATFS-1. The reporter strain has a transgene that links the promoter of the ATFS-1 target gene *hsp-6* to GFP (*Phsp-6::GFP*) such that these worms exhibit increased fluorescence whenever ATFS-1 is activated. After exposing these worms to heat, cold, osmotic stress, anoxia, oxidative stress, starvation, ER stress and bacterial pathogens, we only observed increased fluorescence after exposure to oxidative stress (Dues et al. 2016, *Aging*). Here, we show that constitutive activation of ATFS-1 results in increased resistance not only to oxidative stress but also ER stress, osmotic stress, anoxia and bacterial pathogens (fast kill assay). Thus, ATFS-1 activation does not just protect against stresses that lead to its activation. Notably, the constitutively active *atfs-1* mutants (*et15* and *et17*) exhibit activation of the mitoUPR under unstressed conditions (e.g. upregulation of *hsp-6* in **Fig. 1A**; increased fluorescence of *hsp-6* and *hsp-60* reporter strains in Rauthan et al. 2013, *PNAS*; upregulation of many other stress pathway target genes **Fig. 2**). It is likely that the activation of the mitoUPR and downstream stress response pathways under unstressed conditions results in the increased resistance to stress that we observe. We have included these points in the revised manuscript.

Is there any intervention that controls longevity and does not trigger ATFS-1 response?

When we compared RNA-seq data on a panel of long-lived mutants representing multiple pathways of lifespan extension to ATFS-1 target genes (defined as genes that are upregulated by *spg-7* RNAi in an ATFS-1 dependent manner from Nargund et al. 2012, *Science*), we found that seven of the nine long-lived mutants that we examined showed enrichment of ATFS-1 target genes (*clk-1*, *isp-1*, *nuo-6*, *daf-2*, *glp-1*, *ife-2*) while two did not (*eat-2*, *osm-5*) (**Fig. 5**). Interestingly, in six of these seven strains (all except *ife-2*), there is an increase in reactive oxygen species (ROS) that contributes to their longevity (treatment with antioxidants decreases their lifespan; Yang and Hekimi 2010, *PLoS Biology*; Zarse et al. 2012, *Cell Metabolism*; Wei and Kenyon 2016, *PNAS*). This observation is consistent with the idea that ROS/oxidative stress is sufficient to activate ATFS-1/mitoUPR. We have previously shown that exposure to a mild heat stress (35°C, 2 hours) or osmotic stress (300 mM, 24 hours) can extend lifespan but does

not increase expression of the ATFS-1 target gene *hsp-6* (Dues et al. 2016, *Aging*). Thus, there are multiple examples in which a genetic mutation or intervention increases longevity but does not trigger upregulation of ATFS-1 target genes. We have updated the manuscript to include these points.

3) In Fig. 3, some of these genes appear to be unspecifically associated with different stressors. Therefore, it is difficult to rule out the participation of ATFS-1 in specific stress responses without looking at specific stress-responsive genes or a wider range of genes. For example, the conclusion that ATFS-1 does not control osmotic stress gene expression response comes from looking at 3 genes: *sod-3*, *gst-4* and *Y9C9A.8*. *gst-4* does not appear to be directly controlled by ATFS-1 regardless of the stressor. *sod-3* is also upregulated by oxidative stress and *Y9C9A.8* by anoxia. On the other hand, somewhat contradicting the authors' conclusions that ATFS-1 does not participate in osmotic stress response based on these 3 genes, ATFS-1 appears to be required for osmotic stress resistance.

In this experiment, we treated wild-type and *atfs-1* deletion mutants with six different stressors (oxidative stress, bacterial pathogens, heat stress, osmotic stress, anoxia, and ER stress), isolated mRNA and then examined the expression of 14 different stress response genes. To select these genes, we chose a combination of the most established target genes of the stress response pathways that we examined in Figures 1/2, and genes that we had previously shown to be upregulated by specific stresses using fluorescent reporter strains (Dues et al. 2016, *Aging*). These genes included *hsp-6*, *hsp-4*, *hsp-16.2*, *sod-3*, *gst-4*, *nhr-57*, *Y9C9A.8*, *trx-2*, *ckb-2*, *gcs-1*, *sod-5*, *T24B8.5*, *clec-67* and *dod-22*. To determine if ATFS-1 is required for gene upregulation in response to any of the six different stressors, we first identified which of these stress genes is significantly upregulated in response to each stressor and then looked to see if this upregulation is reduced or prevented by *atfs-1* mutation. We found that there were multiple examples of this for both oxidative stress and bacterial pathogen stress, but not for other stresses. We selected three representative genes to display in **Figure 3**. Nonetheless, it is possible that there are genes that we didn't examine that are upregulated by the other four stressors in an ATFS-1-dependent manner. To definitively address this question, one would have to do RNA sequencing on wild-type and *atfs-1(gk3094)* worms comparing untreated and stressed, but this is beyond the scope of the current manuscript. We have updated the manuscript to include these points, and noted the possibility that there are genes, which we didn't measure, that are upregulated by the other four stressors in an ATFS-1-dependent manner. We have also included the qPCR data for all 14 genes for each of the six external stressors in **Supplemental Figures S3-S8**.

****Minor concerns:****

1) The paragraph starting in line 107 is confusing. They write that "Constitutive activation of ATFS-1 in *atfs-1(et 15)* and *atfs-1(et17)* mutants resulted in upregulation of most of the same genes that are upregulated in *nuo-6* mutants, except for *gst-4*" and later they state that "Activating the mitoUPR through the *nuo-6* mutation, or through the constitutively-active ATFS-1 mutants did not significantly increase the expression of target genes from the ER-UPR (*hsp-4*; Fig. 1B) or the cyto-UPR (*hsp-16.2*; Fig. 1C)." I understand the upregulation of ER-UPR and cyto-UPR is not statistically significant (isn't it for *hsp-16.2*?), but the first sentence is not accurate if statistics is considered.

To clarify this, we have modified the first sentence to describe which genes are significantly upregulated in *atfs-1(et15)* mutants, and separately describe the findings for *atfs-1(et17)* mutants in the second sentence. The results for *hsp-16.2* are not significant because this gene shows highly variable expression between replicates and can be induced 60-fold. We have noted this in the text as well.

2) The authors should discuss why they think *atfs-1(et15)* gain-of-function mutant exhibited decreased resistance to chronic oxidative stress, while it is protected from acute oxidative stress. In fact, the *et15* allele differs in many aspects in relation to the *et17* and in some cases it behaves similarly to the *gk3094* loss-of-function allele.

While *atfs-1(et15)* and *atfs-1(et17)* mutants generally show similar results, they also exhibit differences. We previously used RNA sequencing to examine gene expression in these two strains. We found that *atfs-1(et15)* mutants have far more extensive changes in gene expression than *atfs-1(et17)* mutants (6227 differentially expressed genes versus 958 differentially expressed genes). It is possible that the *et15* mutation is more disruptive to the mitochondrial targeting sequence than *et17*, thereby resulting in increased nuclear localization and more gene expression changes. The additional gene expression changes in the *atfs-1(et15)* mutant may contribute to their decreased resistance to chronic oxidative stress. We have included these points in the revised manuscript.

3) Fig 4I is very similar to Fig. 6A of the other manuscript which strengthen the notion that ATFS-1 is not required (it is rather detrimental) for bacterial pathogen response when no underlying stress (most likely oxidative) occurs.

Yes, our results indicate that ATFS-1 is not required for wild-type survival of bacterial pathogen exposure. This is consistent with our findings in the other manuscript that baseline expression of innate immunity genes does not depend on ATFS-1 (innate immunity gene expression is similar between wild-type and *atfs-1(gk3094)* mutants). We have updated the manuscript to emphasize these points.

4) In the paragraph starting in line 213, the authors conclude that "ATFS-1 is sufficient to protect against oxidative stress, osmotic stress, anoxia, and bacterial pathogens but not heat stress". The results do not unequivocally support a participation of ATFS-1 in oxidative stress or bacterial pathogen response, given the responses vary depending on the allele or condition.

We have modified this sentence by replacing "activation of ATFS-1 is sufficient to protect" with "activation of ATFS-1 can protect" to indicate that we didn't observe protection in all cases.

5) "Combined, this indicates that ATFS-1 does not play a major role in lifespan determination in a wild-type background despite having an important role in stress resistance." It actually does, since ATFS-1 gain-of-function decreases lifespan.

We have rewritten this sentence to say that constitutive activation of ATFS-1 does not extend lifespan, despite increasing resistance to multiple stresses.

6) Paragraph starting in line 359 needs to be discussed in light of the results of the other manuscript submitted by the authors to EMBO.

Combined these two manuscripts indicate that baseline levels of innate immunity are dependent on the p38-mediated innate immune signaling pathway, and not dependent on ATFS-1. This idea is supported by the fact that deletion of *atfs-1* does not decrease resistance to bacterial pathogens and does not reduce the expression of innate immunity genes. In contrast, disrupting genes involved in the p38-mediated innate immune signaling pathway does decrease resistance to bacterial pathogens and does decrease the expression of innate immunity genes. We have updated this paragraph to include these points and reference the findings from our manuscript on innate immunity in the long-lived mitochondrial mutants.

7) In Fig. 1C, it appears that *atfs-1* loss of function increases *hsp-16.2*. Is that significant?

While there is a strong trend towards increased *hsp-16.2* expression in *atfs-1(gk3094)* mutants, this difference did not reach significance because this gene shows highly variable expression and can be induced 60-fold.

8) In Fig. 2, 5 and S1, it would be interesting to build one single Venn Diagram with all the lists of genes to see if there are common genes associated with multiple pathways and if there are many ATFS-1 target genes not associated with these classical stress or longevity pathways.

While we would be very interested in performing this type of visualization, weighted Venn diagrams with more than 3 or 4 groups are challenging to generate and more challenging to interpret. Instead, we have generated an UpSetR plot to demonstrate the number of overlapping genes between each of the stress response pathways, as well as how many ATFS-1 target genes are not involved in stress response. We have included this plot in **Figure 2, Panel I**. We have also generated simpler figure to show the overlap between pairs of stress response pathways (**Figure S1**). In addition, we have also added **Table S4** with these gene lists.

9) In Fig. 2, 5 and S1: What are the p values referred to?

The p-values indicate the significance of the difference between the observed number of overlapping genes between the two gene sets, and the expected number of overlapping genes if the genes were picked at random. We have clarified this in the manuscript.

10) In paragraph starting in line 85, the authors should include references that evidence the genes are bona fide markers of the stress response pathways.

We have added references for each of the genes that we examined to link it to the associated stress response pathway.

11) Tables S2 and S3 are missing.

Tables S2 and **S3** were uploaded as Excel spreadsheets, not included with the supplemental figures as the other supplementary Tables were. We apologize that these were difficult to locate. In the revision, **Table S1** is in the manuscript file, while **Table S2 to S6** will be uploaded as separate files.

Reviewer #2:

****Major comments:****

The only major conclusion that I would qualify is "ATFS-1 serves a vital role in organismal survival of acute stresses through its ability to activate multiple stress response pathways"-the data, as presented, does not make clear whether ATFS-1 directly activates these pathways (ie, by binding response elements in genes in those pathways), or indirectly influences them by altering the physiology of the worm).

We agree that our data does not determine precisely how ATFS-1 acts to modulate the expression of the different stress response pathways. To determine the extent to which ATFS-1 might be able to bind directly to the target genes of other stress response pathways, we have compared the ChIP-seq results for ATFS-1 to ChIP-seq studies for other stress responsive transcription factors (DAF-16, SKN-1, HSF-1, HIF-1, ATF-7). We found that in each case there are sets of genes that can be bound by both transcription factors. This suggests that ATFS-1 may be direct regulating at least some of the target genes from other stress response pathways. We have updated our manuscript to include these points and included the ChIP-seq data comparisons in **Figure S2**.

****Minor comments:****

In abstract, consider broadening/re-wording "Gene expression changes resulting from the activation of the mitoUPR are mediated by the transcription factor ATFS-1/ATF-5." Because a naïve reader may understand this to suggest that ATFS-1 is activated only by mitochondrial protein misfolding.

In this sentence we are describing the role of ATFS-1 in mediating the gene expression changes resulting from the activation of the mitoUPR. We would be happy to modify the sentence if this is unclear.

Please indicate whether strains were outcrossed, and how often.

We have added these details to our materials and methods.

How was "young adult" defined? Were worms synchronized, and if so, how?

Young adult worms are picked on day 1 of adulthood before egg laying begins. The worms were not synchronized, but picked visually as close to the L4-adult transition as possible. We have added these details to our method section.

For the gene expression experiments, do I understand correctly that FUdR was used only for oxidative stress and adult day 2 experiments? Please clarify.

Yes, that is correct. FUdR was used for these samples because (1) with the 2-day duration of this stress, worms can produce progeny which would complicate the collection of the experimental worms; and (2) 4 mM paraquat often results in internal hatching of progeny when FUdR is absent, which might have affected the results. The control worms for the 48-hour 4 mM paraquat stress were also treated with FUdR. We have clarified this in the manuscript and noted that the presence of FUdR has the potential to alter gene expression.

Important: Please make clear how many replicates were performed for each experiment, and where relevant, how many worms were measured per replicate (e.g., stress survival and lifespan).

We have added a spreadsheet (**Table S6**) to include the number of replicates and number of worms per replicate for all experiments.

For 2-way ANOVA analyses, please specify p values of both main factors as well as interaction terms and posthoc analyses where relevant.

We have included these additional details from our statistical analyses in **Table S6**.

In the second paragraph of the introduction, I suggest broadening slightly the description of why normal mitochondrial function is required for ATFS-1 import and degradation, because this helps the reader understand that any one of many perturbations to mitochondrial function (decreased bioenergetics, membrane potential, protein degradation, protein import; increased ROS; etc.) could prevent or reduce ATFS-1 import and degradation.

We have added these additional factors that might prevent ATFS-1 import and degradation in paragraph one of our introduction and broadened the description in paragraph two.

For Figure 1: The authors present their choice of genes to analyze as if, and interpret their results assuming, that each of these gene is ONLY regulated by the indicated stress response pathways. I think this is very unlikely. For example: is it certain that sod-3 and trx-2 are not also skn-1 regulated? How is "antioxidant" distinguished from the skn-1 pathway? Further clouding the water is the likelihood that nuo-6 and atfs-1 manipulations alter physiology in such a way that there are secondary/indirect stress pathways activated (for example: the authors show that ATFS-1 overexpression shortens lifespan. Perhaps this is why it appears that ATFS-1 overexpression also appears to cause a strong, although variable, upregulation of the cytosolic UPR?). The likelihood (in my opinion) that these genes are in fact regulated by more than one type of response element, and that the manipulations used to study these relationships have pleiotropic effects, do not invalidate the general conclusion that these pathways interact-but they do mean that the results should be discussed with more caveats regarding HOW they interact.

These are excellent points. The genes that we selected for **Figure 1** are the genetic targets that in our reading of the literature have been most often used to represent a particular stress response pathway. We have added references to justify the association of each gene with the indicated stress response pathway. We have also noted that in at least some cases the stress response genes that have been typically used to represent a specific pathway can be activated by multiple pathways. We agree that the selection of genes for **Figure 1** is not a comprehensive approach, and that it is possible that if we chose a different gene from each of these pathways, the results might be different. We have updated our manuscript to specifically note these limitations. To avoid these limitations, we examined the overlap between all of the genes significantly upregulated by ATFS-1 activation and all of the genes significantly upregulated by the different stress response pathways in **Figure 2**. In addition, to gain a better understanding of the overlap between these different stress response pathways globally, we have compared gene expression between each of the stress response pathways studied in **Figure S1**.

Figure 1 also illustrates why a more detailed description of sample size and statistical analysis should be provided. What was the "n"? What were the main effects and interaction terms of each 2-way ANOVA? The design is not full factorial and therefore does not permit a simple 2-way ANOVA (i.e., not all condition combinations are performed)-which responses precisely were compared to which? Were 2 2-way ANOVAs performed per mRNA?

For **Figure 1** we used a one-way ANOVA to compare all of the groups to wild-type with a Bonferroni's Multiple Comparison post-hoc test. We have updated the manuscript to include the sample size and statistical details in **Table S6**.

The work shown in Figure 2 is a very nice way to leverage previous data to further explore this idea of cross-talk. I would suggest including a bit more meta-data in the supplemental data files related to each dataset. For example, what lifestages were used (were they all young adult?), was FUDR used, etc.

We have added these details to **Table S3**, which includes the lists of target genes from each stress response pathway.

However, again, I don't understand how the authors can reach this conclusion: "Combined, this indicates that activation of ATFS-1 is sufficient to upregulate genes in multiple stress response pathways." (lines 152-153 but similar phrasing occurs multiple times) Could it not simply be that one form of cellular stress often eventually triggers broader cellular dysfunction, thus activating other cell stress pathways? Ie-how do we know whether these genes are directly regulated by atfs-1 binding regulatory elements, as implied by this phrasing?

This conclusion is derived from our data showing that constitutively active ATFS-1 mutants have significant upregulation of target genes from multiple stress response pathways (**Figure 2**). As the worms in those experiments were not exposed to stress, we don't have reason to believe that they are experiencing cellular stress or dysfunction. We think it is more plausible that activation of ATFS-1, which normally occurs in response to stress, leads to the activation of other stress response pathways, either directly or indirectly, and that these pathways are recruited to help regain mitochondrial homeostasis. We don't mean to imply that activated ATFS-1 binds directly to the target genes of other stress response pathways. We have clarified this in the revised manuscript.

The stress response experiments are very nicely done and very interesting. I appreciate that the authors did not shy away from describing counterintuitive results (eg *et15* mutants showing increased sensitivity to chronic oxidative stress), and think that these results should also be briefly considered in the Discussion.

We have updated our manuscript to discuss the observation that *atfs-1(et15)* mutants have increased sensitivity to chronic oxidative stress.

Figure 3: please report ANOVA interaction terms-these are what tell whether the inductions are in fact dependent on *atfs-1* (not the post-hoc analyses). Again, it also appears that in some cases, there is an upregulation of certain genes with *atfs-1* knockdown-please report all p-values (because there will be many, I recommend a supplemental table with all main and interaction and posthoc analyses). Again, the "n" also needs to be specified.

We have added **Table S6** to include all of these statistical details.

Figure 4 A-C appear to be lacking error bars? Please add. Perhaps relatedly-the effect size for 4A looks much larger than for 4B, but this does not come across in the text.

We have added error bars to Figure 4A-C. We think the difference in effect size might result from the fact that 4A is an acute assay and 4B is a chronic assay. We speculate that the negative effect of the *et15* and *et17* mutations on lifespan might be a stronger factor in the chronic assay. We have updated the text to comment on the relative effect sizes.

For Figures 4 and 6, please indicate sample size-number of independent experimental replicates, and number of worms per replicate (or range per replicate).

We have added the number of replicates and sample size in **Table S6**.

Lines 224-225 re. *sod-2* mutants: these may also act by decreasing ROS signaling (less conversion of superoxide anion to hydrogen peroxide); also, why would this strain not be considered another long-lived mitochondrial mutant (like *clk-1*, *isp-1* and *nuo-6*, to which it is contrasted)?

We think the *sod-2* mutation extends lifespan by increasing ROS signaling, as treatment with antioxidants decreases their lifespan. The increased superoxide from the loss of *sod-2* may be converted to H₂O₂ by *sod-3* or *sod-1*, which are also present in the mitochondria. We don't include *sod-2* with the mitochondrial mutants because the mutation does not directly impact the mitochondrial electron transport chain, but may do so secondarily due to elevated ROS.

The confirmation that *atfs-1* overexpressing strains are short-lived is very interesting. However, I think this statement "Combined, this indicates that ATFS-1 does not play a major role in lifespan determination in a wild-type background despite having an important role in stress resistance." (lines 265-267 and similar in several places throughout the Discussion, eg line 279) should be altered to indicate that this was observed under controlled laboratory conditions. Eg, "...this indicates that ATFS-1 does not play a major role in lifespan determination in a wild-type background under optimized laboratory conditions..."

This is an interesting point. It is possible that constitutive activation of ATFS-1 may be beneficial for lifespan in an environment where worms are exposed to external stressors. We have noted that our lifespan results were obtained under lab conditions, which are believed to be relatively unstressful.

Discussion: consider adding in a consideration of dose-response, both of knockdown of mitochondrial genes (eg, k/d of many mitochondrial genes promotes lifespan at low levels, but decreases lifespan with greater knockdown) and of stressors (chemicals, heat, etc; for chemicals, at the least, dose-response is very important, with low levels not infrequently triggering apparently beneficial stress responses, and higher levels causing toxicity).

It is possible that the magnitude of ATFS-1 activation will impact its effect on stress resistance and lifespan. Perhaps, a milder activation of ATFS-1 will be more beneficial with respect to lifespan. The degree of ATFS-1 activation may also account for differences that we observe between *atfs-1(et15)* and *atfs-1(et17)* mutants. *atfs-1(et15)* has more differentially expressed genes than *atfs-1(et17)* suggesting the possibility that it has more ATFS-1 activation. We have updated our manuscript to include these points.

Section beginning on line 384 "ATFS-1 upregulates target genes of multiple stress response pathways"-again, please revise to make clear that this work does not demonstrate direct regulation.

We have clarified that our results don't demonstrate direct regulation. In addition, we have examined published ChIP-seq datasets to determine if there is evidence of direct regulation.

It seems to me that our reviews are in pretty good agreement. I agree with Reviewers 1 and 3 where they commented on things that I did not. While I did not consider the manuscripts as overlapping in the sense of being redundant, I very much like Reviewer 1's suggestion that they be published back to back and that the Discussion of each incorporate consideration of the Results of the other.

According to this suggestion, we have arranged for these papers to be considered for publication at the same time in *EMBO Reports* and *Life Science Alliance*. We have updated the discussions of both manuscripts to incorporate the findings of the other manuscript.

Reviewer #3:

****Major comments****

1. The authors mention that activation of the UPRmt by *nuo-6* mutants or *atfs-1(gf)* do not activate the ER UPR or cyto-UPR gene expression targets (lines 111-113). However, they also find that *atfs-1(gf)* animals have 25% overlap with the ER UPR pathway (line 146-147). Is 25% overlap not substantial?

The genes that we are referring to in lines 111-113 are the genetic targets that in our reading of the literature have been most often used to represent the ER-UPR or Cyto-UPR. This is not a comprehensive approach, and it is possible that if we chose a different gene from each of these pathways, the result might be different. We have updated our manuscript to include this limitation. To avoid this limitation, we examined the overlap between all of the genes significantly upregulated by ATFS-1 activation and all of the genes significantly upregulated by the ER-UPR or Cyto-UPR in **Figure 2**. In both cases, we find the overlap is significant, indicating that activation of ATFS-1 leads to activation of ER-UPR and Cyto-UPR target genes.

To determine whether ATFS-1 mediates any protective effect during ER stress, authors should test *atfs-1(gf)* and *atfs-1(lf)* animals' resistance to ER stress.

To examine the effect of ATFS-1 on resistance to ER stress, we exposed wild-type, *atfs-1(gk3094)*, *atfs-1(et15)* and *atfs-1(et17)* worms to 50 μ M tunicamycin beginning at young adulthood and monitor survival daily. We found that both constitutively active *atfs-1* mutants, *et15* and *et17*, have increased resistance to ER stress compared to wild-type worms, while *atfs-1* deletion mutants have a similar survival to wild-type. We have added this new data to **Figure 4**.

2. Authors should comment on the difference in outcomes with *atfs-1(et17)* and *atfs-1(et15)* animals to chronic oxidative stress (line 184-187).

We have updated our manuscript to discuss the observation that *atfs-1(et15)* mutants have increased sensitivity to chronic oxidative stress.

3. Lines 258-260. The authors should make clear in this section that a previous study had already measured lifespans of *atfs-1(gf)* animals and found that it was reduced (PMID 24662282). Also, an elaboration on why this experiment was repeated would be warranted.

We have referenced the lifespan results from this previous study in our introduction (line 53-54, Bennett et al), in our results section (lines 342-343; “which is consistent with a previous study finding shortened lifespan in *atfs-1(et17)* and *atfs-1(et18)* worms”) and in our discussion (lines 429-431; “as well as previous results using constitutively active *atfs-1* mutants (*et17* and *et18*) show that constitutive activation of ATFS-1 in wild-type worms results in decreased lifespan”). The reasons that we repeated this result are (1) because the lifespan of the *atfs-1(et15)* mutant had not been measured and this was the allele that we used in our paper; and (2) because the shortened lifespan is a surprising result given the beneficial effect of ATFS-1 on stress resistance, we thought it was important to repeat this experiment under the same conditions that we measured stress resistance.

4. The authors find that *atfs-1(gk3094)* animals lived longer during infection with PA14 (line 208-211). Another study found that *atfs-1(gk3094)* animals died faster on PA14 (PMID 28283579), which should be mentioned and commented on.

We have added this finding to our discussion. We have also compared the protocols used by Jeong et al. (who observed decreased survival in *atfs-1(gk3094)* deletion mutants), Pellegrino et al. (who observed wild-type survival in *atfs-1(tm4919)* deletion mutants and our manuscript (in which we observed slightly increased survival in *atfs-1(gk3094)* deletion mutants), to see which parameters might account for the observed differences.

****Minor comments****

Line 38: "Inside the mitochondria, ATFS-1 is degraded by the Lon protease CLPP-1/CLP1". The phrasing suggests that CLPP-1/CLP1 is a Lon protease, when in fact they are independent proteases.

We have removed the word “Lon” to clarify this.

August 18, 2021

RE: Life Science Alliance Manuscript #LSA-2021-01182

Jeremy Van Raamsdonk

Dear Dr. Van Raamsdonk,

Thank you for submitting your revised manuscript entitled "Activation of mitochondrial unfolded protein response protects against multiple exogenous stressors". We would be happy to publish your paper in Life Science Alliance pending final revisions necessary to meet our formatting guidelines.

- please add a Running Title to our system
- please add a Summary Blurb/Alternate Abstract in our system
- please add a Category for your manuscript in our system
- please add the Twitter handle of your host institute/organization as well as your own or/and one of the authors in our system
- please add Contributions of all Authors in our system
- please consult our manuscript preparation guidelines <https://www.life-science-alliance.org/manuscript-prep> and make sure your manuscript sections are in the correct order
- we encourage you to revise the figure legend for figure 1 such that the figure panels are introduced in an alphabetical order
- please add callouts for Figure S1A, B to your main manuscript text

LSA now encourages authors to provide a 30-60 second video where the study is briefly explained. We will use these videos on social media to promote the published paper and the presenting author. Corresponding or first-authors are welcome to submit the video. Please submit only one video per manuscript. The video can be emailed to contact@life-science-alliance.org

A. FINAL FILES:

B. MANUSCRIPT ORGANIZATION AND FORMATTING:

Sincerely,

Reviewer #1 (Comments to the Authors (Required)):

The authors have satisfactorily addressed my concerns and I believe the manuscript can now be accepted for publication.

Reviewer #2 (Comments to the Authors (Required)):

The authors have addressed my concerns satisfactorily.

Reviewer #3 (Comments to the Authors (Required)):

The authors have addressed all the concerns raised in my initial review. I am satisfied with the current state of the manuscript and support its acceptance.

September 20, 2021

RE: Life Science Alliance Manuscript #LSA-2021-01182R

Prof. Jeremy Michael Van Raamsdonk
McGill University
Neurology and Neurosurgery
1001 Decarie Boulevard
Montreal, QC H4A 3J1
Canada

Dear Dr. Van Raamsdonk,

Thank you for submitting your Research Article entitled "Activation of mitochondrial unfolded protein response protects against multiple exogenous stressors". It is a pleasure to let you know that your manuscript is now accepted for publication in Life Science Alliance. Congratulations on this interesting work.

DISTRIBUTION OF MATERIALS:

Again, congratulations on a very nice paper. I hope you found the review process to be constructive and are pleased with how the manuscript was handled editorially. We look forward to future exciting submissions from your lab.

Sincerely,
